# Holocene shifts in marine mammal distributions around Northern Greenland revealed by sedimentary ancient DNA

Lennart Schreiber [1,2] ✉, Sofia Ribeiro [1,2,7] ✉, Rebecca Jackson [3], Anna Bang Kvorning [1,2], Kevin Nota[4], Matt O'Regan [5], Christof Pearce [6], Frederik Seersholm [1], Marit-Solveig Seidenkrantz [6], Heike H. Zimmermann [2] & Eline D. Lorenzen [1,7] ✉

Arctic marine ecosystems have undergone notable reconfigurations in response to Holocene climate and environmental changes. Yet our understanding of how marine mammal occurrence was impacted remains limited, due to their relative scarcity in the fossil record. We reconstruct the occurrence of marine mammals across the past 12,000 years through detections based on sedimentary ancient DNA from four marine sediment cores collected around Northern Greenland, and integrate the findings with local and regional environmental proxy records. Our findings indicate a close association between marine mammals at densities detectable in marine sediments and the deglaciation of high Arctic marine environments at the onset of the Holocene. Further, we identify air temperature and changes in sea ice cover as significant drivers of community change across time. Several marine mammals are detected in the sediments earlier than in the fossil record, for some species by several thousand years. During the Early-to-Mid Holocene, a period of warmer climate, we record northward distribution shifts of temperate and low-arctic marine mammal species. Our findings provide unique, long-term baseline data on the occurrence of marine mammals around Northern Greenland, enabling insights into past community dynamics and the effects of Holocene climatic shifts on the region's marine ecosystems.

Ongoing anthropogenic climate change is transforming the Arctic. The region is warming four times faster than the global average[1] and the extent of summer sea ice has already declined by as much as 50% over the past 40 years[2]. Temporarily ice-free conditions in the Arctic Ocean, defined as the first occurrence of a total sea ice area < 1 million km², may already emerge during summer by the end of this decade[3]. In response to these shifting environmental conditions, the spatial distribution of marine species across the Arctic is changing[4–7]. Species endemic to the Arctic region are expected to face a drastic reduction in suitable habitats[8,9], while temperate species may benefit from the reconfiguration of oceanic currents and increased availability of ice-free habitats[10,11].

Arctic marine mammals are sensitive to climate change, especially due to their dependence on sea ice[12,13]. Seven species are found around

[1]Globe Institute, University of Copenhagen, Copenhagen, Denmark. [2]Department of Glaciology and Climate, Geological Survey of Denmark and Greenland, Copenhagen, Denmark. [3]MARUM – Center for Marine Environmental Sciences, Bremen, Germany. [4]Department of Evolutionary Genetics, Max Planck Institute for Evolutionary Anthropology, Leipzig, Germany. [5]Department of Geological Sciences, Stockholm University, Stockholm, Sweden. [6]Department of Geoscience, Arctic Research Center, and iClimate Center, Aarhus University, Aarhus, Denmark. [7]These authors contributed equally: Sofia Ribeiro, Eline D. Lorenzen. ✉e-mail: lennart.schreiber@sund.ku.dk; sri@geus.dk; elinelorenzen@sund.ku.dk

Northern Greenland year-round, including the three Arctic cetacean species [beluga whale (*Delphinapterus leucas*), narwhal (*Monodon monoceros*), and bowhead whale (*Balaena mysticetus*)], three pinniped species [ringed seal (*Pusa hispida*), bearded seal (*Erignathus barbatus*), walrus (*Odobenus rosmarus*)], and polar bear (*Ursus maritimus*). In addition, harp seal (*Phoca groenlandica*) and hooded seal (*Cystophora cristata*) seasonally migrate northwards along the east and west coasts of Greenland, and depend on the presence of sea ice for reproduction[14,15].

At least 12 temperate cetacean species with distribution ranges not confined to the Arctic have been recorded in Greenland, with species such as fin whale (*Balaenoptera physalus*) and minke whale (*Balaenoptera acutorostrata*) seasonally migrating in large numbers to ice-free feeding grounds along the central and northern coasts of Greenland[16,17]. The observed impacts of ongoing climate change on marine mammals found around Greenland are not uniform[18]. For example, the extended open-water season may boost the availability of prey (e.g., krill, copepods) for species such as bowhead whales[19,20], but the associated increase in the presence of orcas (*Orcinus orca*) also poses higher risks of predation[21].

Monitoring efforts provide crucial data on population dynamics, distribution shifts, and behavioral changes in response to changing environmental conditions[13,22,23]. However, we lack an understanding of the longer-term dynamics of Arctic marine ecosystems. Integrating knowledge on the shorter-term response of species to anthropogenic climate change[24] with longer-term perspectives on how species distributions have shifted in response to past environmental changes may enhance our understanding of their resilience to global warming[25].

Population genetic studies of marine mammals in the Arctic document the impacts of the environmental changes of the Holocene (past 11.7 thousand years [ka]) on present-day population structure[26–28]. Holocene paleoclimate records reveal periods with regional air temperatures up to 6 °C warmer than the pre-industrial average[29], which may be used as near-future climate analogs. Specifically, a period in the Early-to-Mid Holocene was characterized by elevated air and ocean temperatures, as a consequence of higher boreal summer insolation, known as the Holocene Thermal Maximum (here broadly used to refer to the period of peak Holocene warmth ~10–5.5 ka BP)[30]. As a result, marine outlet glaciers along the coasts of Greenland retreated dramatically[31–33], sea ice extent and thickness decreased as indicated by driftwood deposition along the northern coast of Ellesmere Island and Greenland[34,35], and oceanic currents such as the West Greenland Current (Fig. 1a) intensified[36].

Information on the past occurrence of species can be retrieved from the fossil record. However, discoveries of marine mammal fossils are rare, as most species live and die at sea. Archeological sites are rich sources of faunal remains, but are limited to the species harvested by humans[37] and for Greenland extend back only 4.5 ka[38]. Although individual, older remains of e.g., bowhead whale, narwhal, ringed seal, walrus, and polar bear, have been recovered from paleontological contexts[39–42], a detailed, long-term reconstruction of these species' histories is limited by the scarcity of fossils.

Sediment archives provide an alternative substrate to document the occurrence of species through time[43]. In the terrestrial realm, sediment archives have been used to broaden our understanding of spatiotemporal patterns of species occurrence. For example, DNA retrieved from permafrost sediments has indicated the presence of species in the landscape for millennia after they were presumed extinct based on their last appearance dates in the fossil record[44,45].

Marine sediments have long been used for paleoenvironmental reconstruction[46], but their full potential for detecting species occurrence using DNA remains underexplored[47,48]. So far, most research on marine sediments has been based on traditional micropalaeontological methods and has focused on obtaining time-series data from organisms at the base of the food web[49–53]. However, the retrieval of sedimentary ancient DNA from higher trophic level organisms provides the opportunity to improve our understanding of the dynamics of marine ecosystems through time[54,55].

In this work, we characterize the occurrence of marine mammal species across the Holocene, using sedimentary ancient DNA retrieved from four marine sediment cores collected around the northern coast of Greenland that span the past ca. 12 ka (Fig. 1). We applied two complementary molecular approaches (metagenomic shotgun sequencing and hybridization capture enrichment of mitochondrial genomes) for marine mammal detection, and integrated our findings with paleoceanographic proxy data derived from the same or nearby sediment cores that provide information on primary productivity, sea ice cover, and the influence of Atlantic-derived (warmer) water masses. Specifically, we investigated (i) the occurrence through time of the seven marine mammals that at present inhabit the Atlantic sector of the Arctic year-round; (ii) the occurrence through time of marine mammal species that at present only inhabit the Arctic seasonally; and (iii) spatiotemporal associations between marine mammal occurrence and Holocene paleoenvironmental proxy data. Our findings indicate environmental changes as significant drivers of community shifts across time; we identify the presence of marine mammals at densities detectable in marine environments coincident with Arctic marine deglaciation at the onset of the Holocene, and record subsequent northward distribution shifts of temperate and low-arctic marine mammal species during the warmer Early-to-Mid Holocene.

## Results
### Overview of sequencing data and DNA damage
Using a combination of shotgun sequencing and mitochondrial genome hybridization capture, we retrieved DNA time-series data spanning the past ~12 ka from four marine sediment cores collected off the coasts of Northern Greenland, albeit with varying temporal resolution (Fig. 1); LK21-IC-st26-GC1 (75.32° N 61.91° W, 912 m water depth, 320 cm sediment recovery; hereafter Melville Bay 26G), Ryder19-24-PC1 (81.62° N 62.30° W, 520 m water depth, 525 cm sediment recovery; hereafter Hall Basin 24PC), Ryder19-12-GC1 (82.58° N 52.53° W, 867 m water depth, 273.5 cm sediment recovery; hereafter Lincoln Sea 12-GC) and DA17-NG-ST07-073G (79.07° N 11.90° W, 385 m water depth, 410 cm sediment recovery; hereafter North-East Greenland 73G). Melville Bay 26G and Lincoln Sea 12-GC comprise a complete record of the Holocene, while Hall Basin 24PC covered the past ca. 9.6 cal ka BP and North-East Greenland 73G covered the past ca. 9.5 cal ka BP (Fig. 1c).

Subsamples of 0.35 - 1 g of sediment were collected from each of the cores (11–42 samples per core) for DNA analysis, resulting in 116 samples, of which all were processed using hybridization capture, and a subset was subjected to additional shallow shotgun sequencing (Fig. 1c, and Supplementary Data 1). When comparing DNA concentrations of total DNA extracts across sediment cores and samples, samples < 2 ka BP generally had the highest DNA concentrations ($7.4 \pm 5.1$ ng μL$^{-1}$), while older samples yielded lower concentrations ($1.4 \pm 1.5$ ng μL$^{-1}$) or remained below the detection limit of 0.05 ng μL$^{-1}$ (Figs. 2d, 3d, 3g, 4d).

We used shotgun sequencing to characterize the most abundant eukaryotic taxa and hybridization capture to specifically enrich the genetic traces of marine mammals endemic to the Arctic and of relevant marine mammals, whose present-day distribution is not confined only to the Arctic. We investigated the association of the combined dataset with paleoceanographic proxy data to identify potential drivers of change through time. All genetic detections are reported per taxon and sample, in unique sequence abundances (defined as the number of unique DNA sequences) and in relative sequence abundances (defined as the number of unique sequences divided by the number of quality-filtered sequences). Both metrics are interpreted as confidence of the genetic signal underlying the detection of a taxon; high unique sequence abundance and high relative sequence abundance translate into a detection with higher confidence, and low

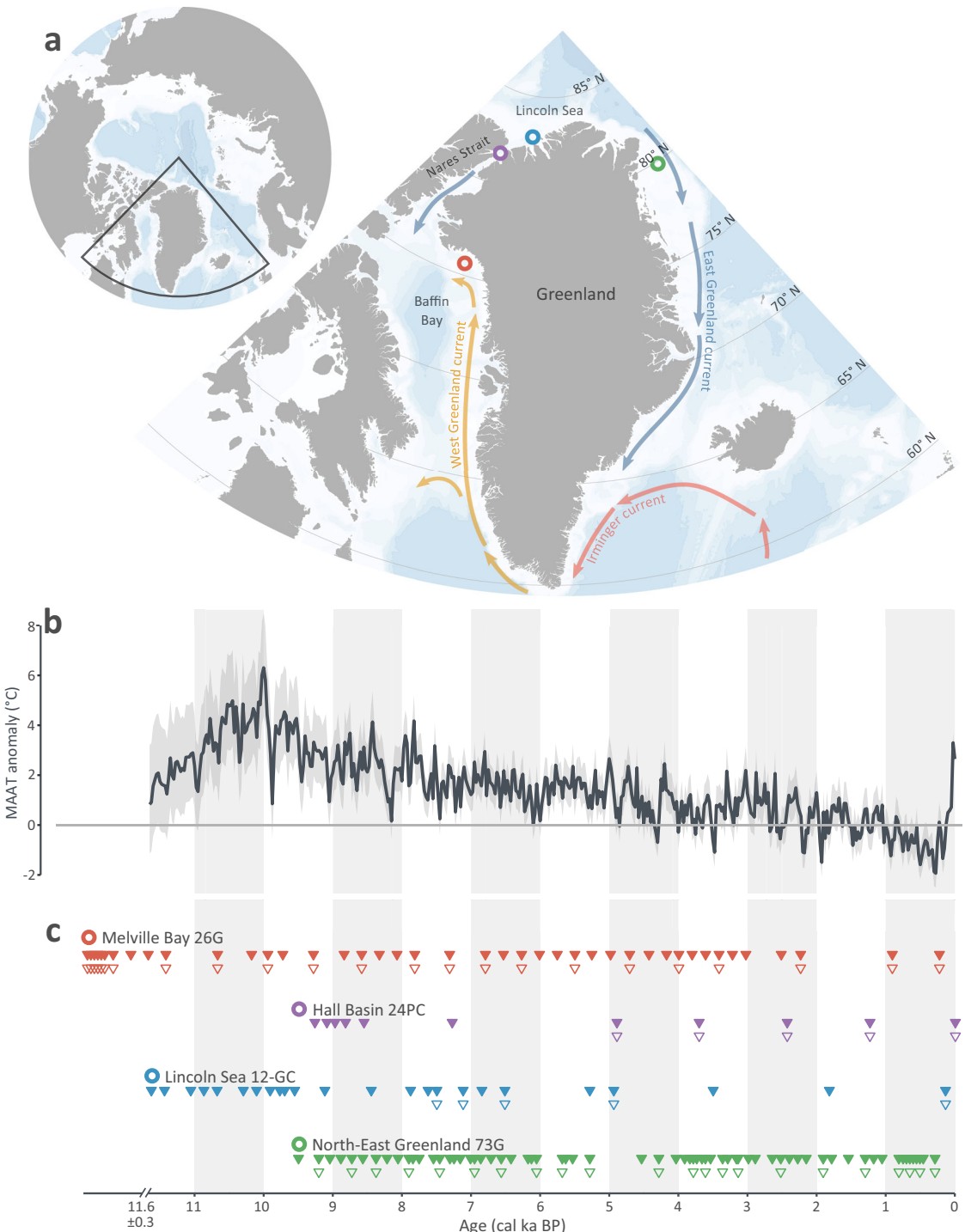

**Fig. 1 | Sites and samples of the four marine sediment cores analyzed. a** Map of Greenland showing the localities of the cores, indicated by colored circles: Melville Bay 26G (red; 75° N; LK21-IC-st26-GC1); Hall Basin 24PC (purple; 82° N; Ryder19-24-PC1); Lincoln Sea 12-GC (blue; 83° N; Ryder19-12-GC1); North-East Greenland 73G (green; 79° N; DA17-NG-ST07-073G). Data from NOAA National Centers for Environmental Information, U.S. Department of Commerce, distributed as a U.S. Government Work (public domain). **b** Holocene air temperature reconstruction based on the oxygen isotopic signature ($\delta^{18}O$) at the Agassiz ice cap, as compared to the pre-industrial (1850-1900 CE) average[29,56]. **c** Each sediment core was subsampled for DNA across its length. Triangles indicate the estimated median age of each sample based on the age-depth models of each core (Melville Bay 26G: this study; Hall Basin 24PC: this study; Lincoln Sea 12-GC[56]; North East Greenland 73G[57]); filled triangles indicate samples processed with hybridization capture; empty triangles indicate samples subjected to shotgun sequencing.

unique sequence abundance and low relative sequence abundance into a detection with lower confidence.

The analysis of nucleotide deamination rates (DNA damage) in the shotgun sequencing and hybridization capture data enabled us to identify the accumulation of nucleotide deaminations for older samples typical of ancient DNA, and thus provided an authentication of our species detections. We visualized mean nucleotide deamination rates across the last three bases of sequences assigned to eukaryotes for the shotgun sequencing dataset, and sequences assigned to marine mammals in the hybridization capture dataset for each core

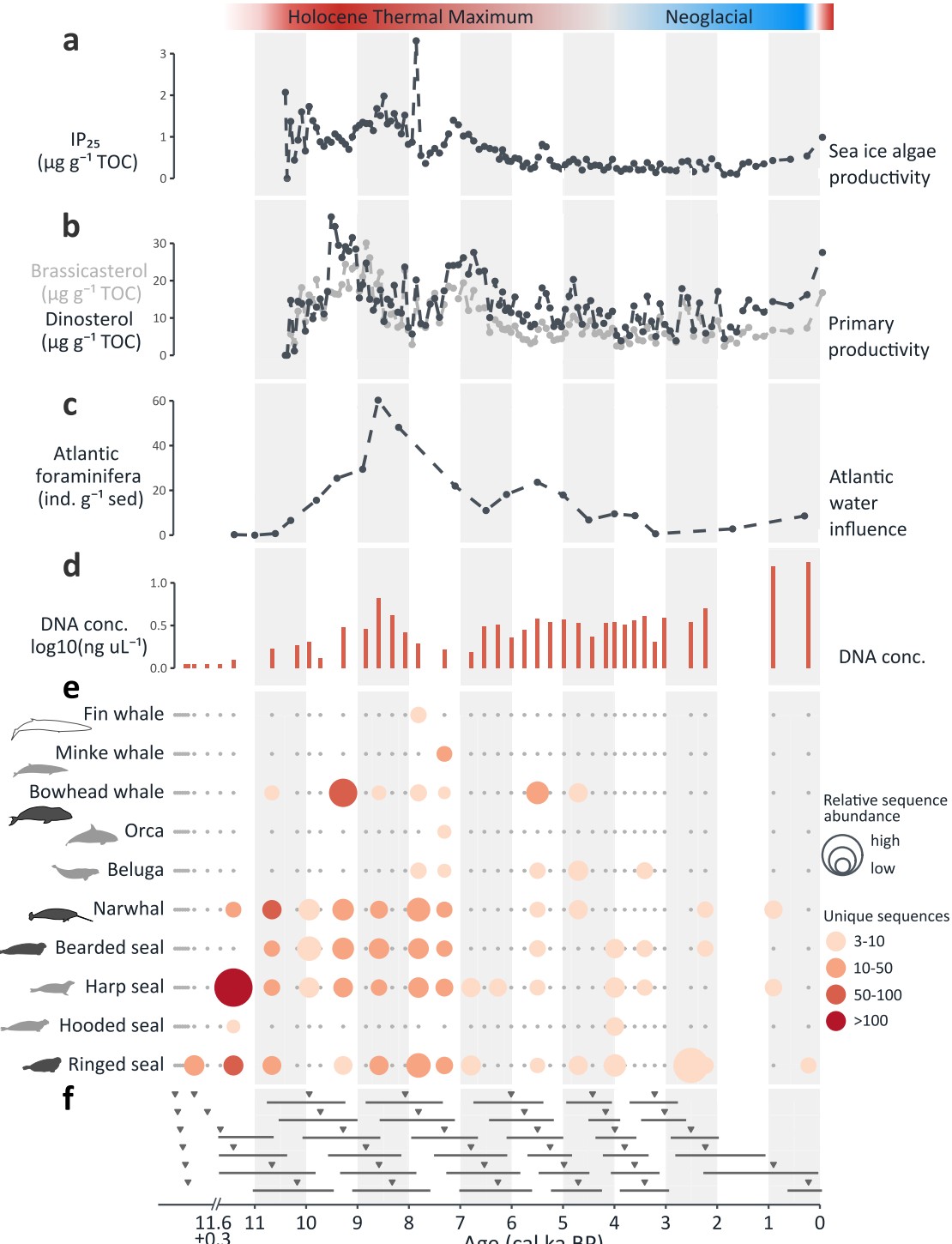

**Fig. 2 | Paleoceanographic reconstructions and DNA detections for marine sediment core Melville Bay 26G. a** Sea-ice biomarker $IP_{25}$ and (**b**) primary-productivity biomarkers brassicasterol and dinosterol are derived from marine sediment core GeoB19927-3[76]. **c** High numbers of benthic foraminifera with preferred habitat of Atlantic-sourced waters indicate the influence of chilled Atlantic water, points represent counts per gram sediment (this study). **d** DNA concentrations of the DNA extracts of the sediment samples. **e** Marine mammal detections through hybridization capture. Silhouettes are shade-coded by present-day year-round occurrence (black), seasonal presence (gray), and absence (white) in the region; silhouettes were drawn by Anna Bang Kvorning. **f** Triangles show the estimated median age of each sample, with horizontal lines indicating confidence intervals. Samples with estimated median ages >11.6 cal ka BP likely represent a short time period at or just prior to the deglaciation of Melville Bay (estimated 11.6 ± 0.3 ka BP[63]).

(Supplementary Fig. 1). Overall, elevated mean nucleotide deamination rates were detected for most samples >3 cal ka BP, especially for taxa with >100 sequences. Using full nucleotide deamination rate, fragment length distribution and coverage profiles of all DNA sequences

assigned to Phocidae across time bins of 3000 years for each core (Supplementary Figs. 2–5), we observed higher nucleotide deamination rates for DNA sequences corresponding to older samples and overall low DNA fragment lengths (≤125 bp).

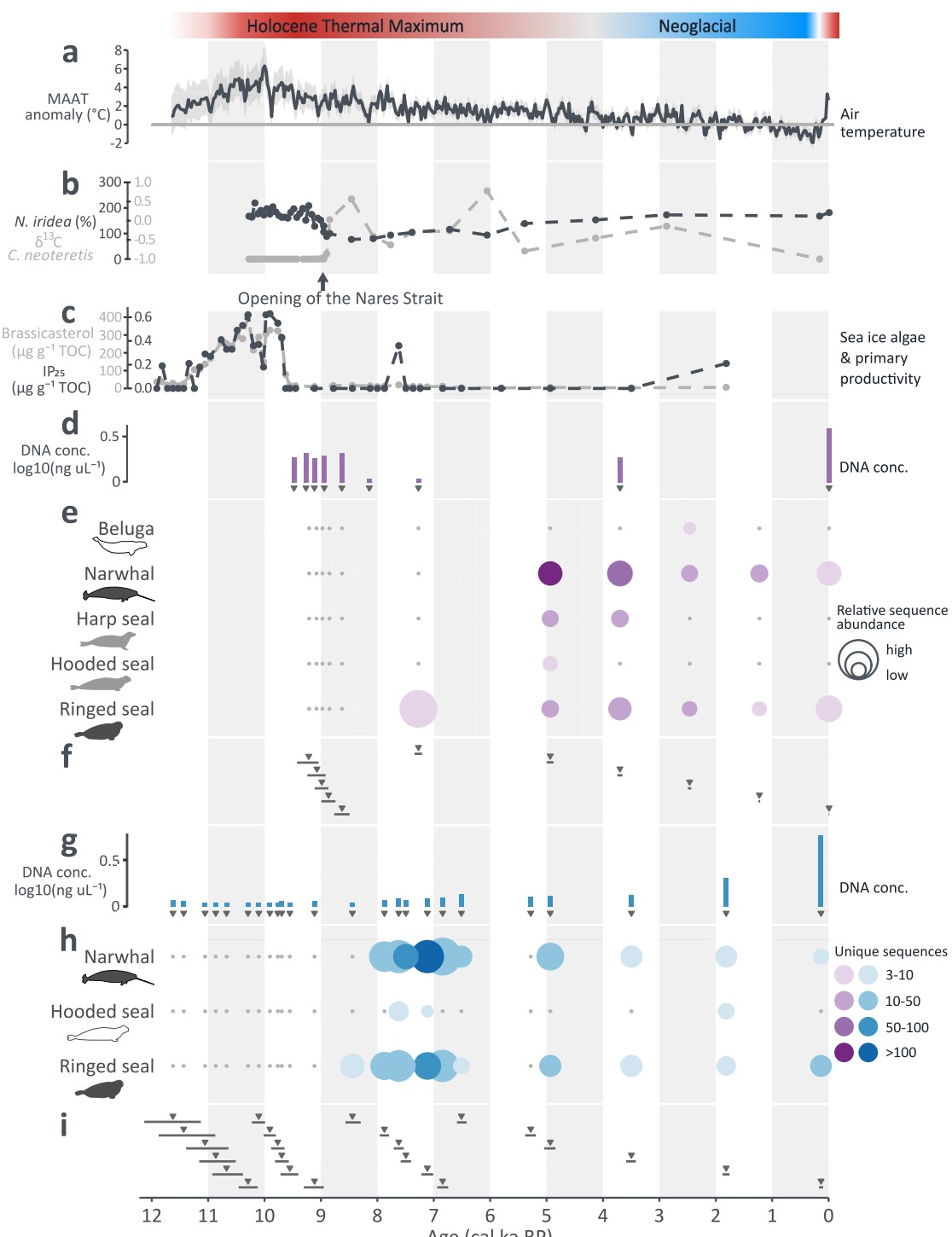

**Fig. 3 | Palaeoenvironmental reconstructions and DNA detections for marine sediment cores Hall Basin 24PC and Lincoln Sea 12-GC. a** Temperature reconstruction based on the oxygen isotopic signature (δ18O) at the Agassiz ice cap as compared to the pre-industrial (1850-1900 CE) average[29,56]. **b** Evidence for the opening of Nares Strait (derived from core HLY03-01-05GC[51]). **c** Sea-ice biomarker IP25 and primary-productivity biomarker brassicasterol, derived from Lincoln Sea 12-GC[56]. **d–i** DNA data from (**d–f**) Hall Basin 24 C are shown in purple; corresponding data from (**g–i**) Lincoln Sea 12-GC are shown in blue. **d**, **g** DNA concentrations of the DNA extracts of the sediment samples; **e/h** Marine mammal detections through hybridization capture; **f, i** Triangles show the estimated median age of each DNA sample, with horizontal lines indicating confidence intervals. Silhouettes are shade-coded by present-day year-round occurrence (black), seasonal presence (gray), and absence (white) in the region; silhouettes were drawn by Anna Bang Kvorning.

## Age-depth models of the studied marine sediment cores

We established age-depth models for the Melville Bay 26G and Hall Basin 24PC cores. Age-depth models are available for Lincoln Sea 12-GC (Supplementary Fig. 6)[56] and North-East Greenland 73G (Supplementary Fig. 7)[57].

For Melville Bay 26G, we constructed an age-depth model using 14C radiocarbon dates of three mixed planktonic (2.7 ± 0.3 cal ka BP; 4.2 ± 0.3 cal ka BP; 5.3 ± 0.4 cal ka BP) and three mixed benthic (7.3 ± 0.7 cal ka BP; 8.8 ± 0.7 cal ka BP; 10.1 ± 0.8 cal ka BP) foraminifera samples (Supplementary Data 2), 210Pb excess activity from the

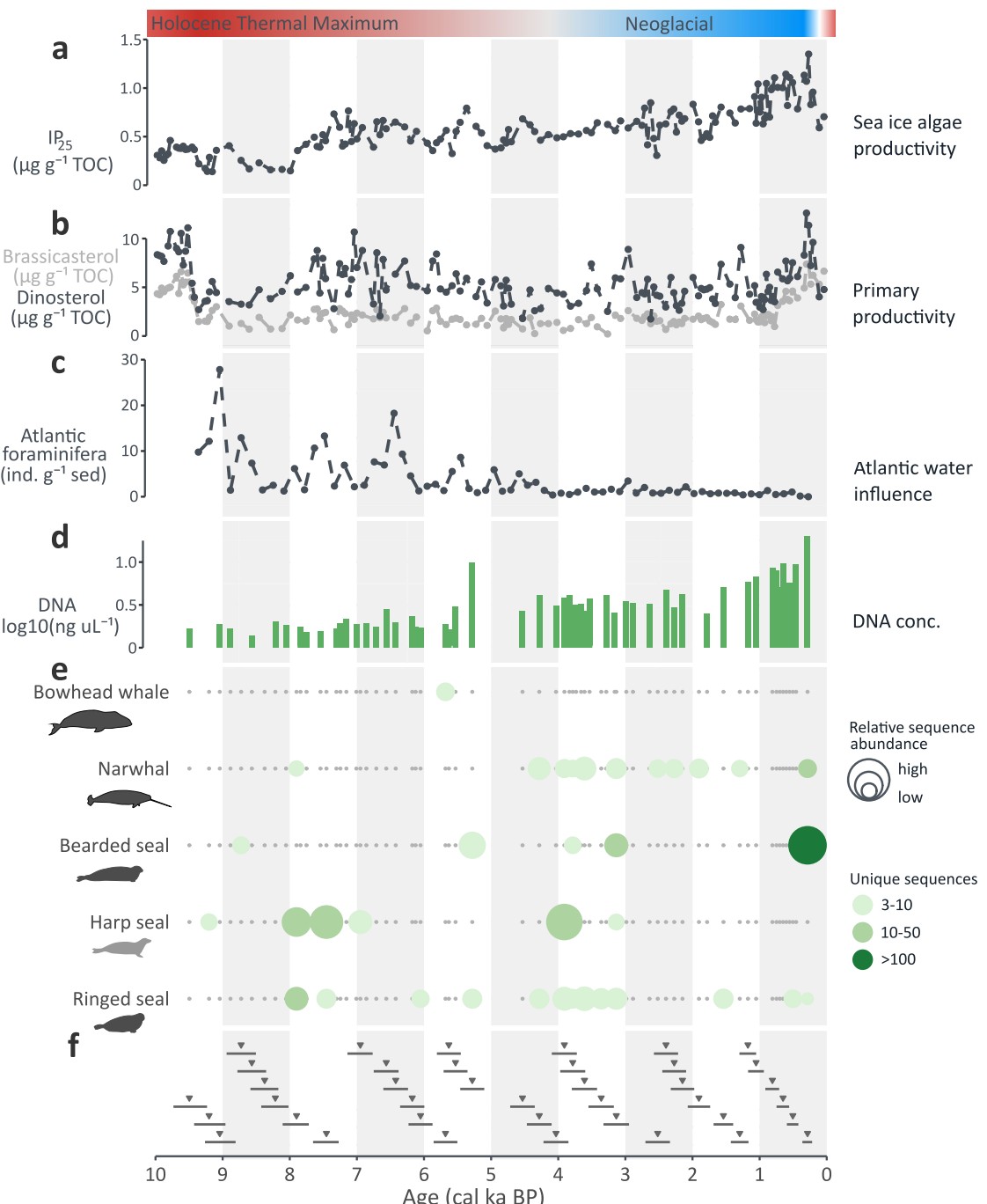

**Fig. 4 | Paleoceanographic reconstructions and DNA detections for marine sediment core North-East Greenland 73G. a** Sea-ice biomarker IP$_{25}$ and (**b**) primary productivity biomarkers brassicasterol and dinosterol are derived from marine sediment core PS93/025-2[99]. **c** High numbers of benthic foraminifera with preferred habitat of Atlantic-sourced waters indicate the influence of Atlantic water, points represent counts per gram sediment (derived from marine sediment core North-East Greenland 73G[57]). **d** DNA concentrations of the DNA extracts of the sediment samples. **e** Marine mammal detections through hybridization capture. Silhouettes are shade-coded by present-day year-round occurrence (black), seasonal presence (gray), and absence (white) in the region; silhouettes were drawn by Anna Bang Kvorning. **f** Triangles show the estimated median age of each DNA sample, with horizontal lines indicating confidence intervals.

uppermost part of the sediment core (Supplementary Data 3), and an upper age constraint for the surface of the sediment core corresponding to the year of retrieval (2021 CE) (Supplementary Fig. 8). The lower part of Melville Bay 26G (320 - 213 cm) was laminated with high magnetic susceptibility and we did not find sufficient microfossils for [14]C dating in this part of the core (Supplementary Fig. 8). At 213 cm core depth, we noted a shift in the sedimentology from laminated to homogenous and a pronounced increase in the magnetic susceptibility, followed by a gradual decrease towards the surface of the core.

Due to the observed shift in sedimentology and the absence of datable material in the lower part of the core, we extrapolated the age-depth model only up to the depth where the sedimentology changes (213 cm depth; 12.3 ± 1.4 cal ka BP). The upper section of Melville Bay 26G (213−0 cm) was homogeneous with larger drop stones deposited throughout. Due to lower sedimentation rates in the uppermost part of the sediment core (14−0 cm), the temporal resolution of the past two thousand years is limited to two samples (Fig. 1). The age-depth model was constrained in the top of the sediment core to the year of retrieval

(2021 CE). A modern age for the surface of the core is supported by the presence of detectable [210]Pb excess activity in the uppermost centimeter of the core (Supplementary Data 3). Raw and calibrated [14]C dates are listed in Supplementary Data 2 and Supplementary Fig. 8 shows an overview of the final chronology.

Hall Basin 24PC was positioned to re-core a site first sampled during the Petermann 2015 expedition of *I/B Oden* (OD1507-18GC[58]; 81.63° N, 62.30° W; 4.35 m long). The two coring sites are less than 450 m apart, and an integrated stratigraphy and age model for OD1507-18GC/Ryder19-24PC has been published[58]. The radiocarbon dates from OD1507-18GC (four [14]C dates performed on monospecific aliquots of the benthic foraminifera *Cassidulina neoteretis*, and a single paired date of the planktic foraminifera *Neogloboquadrina pachyderma*) were migrated onto the depth scale of Hall Basin 24PC (Supplementary Fig. 9, Supplementary Data 4). The age-depth model was constrained by assuming the top of the sediment core corresponded to the year of collection (2019 CE), and a further age constraint was applied at the transition into the basal diamicton. Here we assigned an age of 8910 ± 100 BP, based on the interpreted timing of the retreat of glacial ice from the Polaris Foreland region, near to where the core was collected[59,60].

Hall Basin 24PC is composed of three major lithofacies (Supplementary Fig. 9 and S10). An upper bioturbated mud with clasts (0-134.5 cm) divided into an IRD-rich bioturbated mud (BI) (0-46 cm) and IRD-poor bioturbated mud (BM) (46-134.5 cm). The bioturbated muds are separated from an underlying laminated mud (LM) by an IRD-rich layer (I) (134.5-164.5 cm). The laminated mud (164.5-525 cm) transitions sharply into a structureless diamicton (Dm) (5.25-672.5 cm) interpreted as an ice-proximal facies[58] (Supplementary Fig. 9).

### Shotgun sequencing data
After quality-filtering the shotgun sequencing data, a total of 478–109,137 sequences per sample (median: 4548) were taxonomically assigned (Supplementary Data 1). Of these, 64–100% (median: 96%) were assigned to prokaryotes, which were excluded from further analysis. All ten blanks as well as five sediment samples did not contain any taxonomic assignments after quality-filtering. The resulting taxonomic profile contained only six eukaryotic families (or ten, if eukaryotic families detected in less than three samples are considered), as expected from the relatively shallow sequencing depth of 1.5 ± 0.7 M raw sequences per sample. We refrained from deeper shotgun sequencing due to the high cost of data needed to exhaustively describe samples.

Our shotgun sequencing data revealed DNA of codfishes (Gadidae) and pinnipeds (Phocidae) in all four sediment cores, and DNA of diatoms (Chaetoceraceae) in all cores except for North-East Greenland 73G (Supplementary Data 1, Supplementary Fig. 11–S13). Across sediment cores, we detected a higher number of eukaryotic families in samples >2 cal ka BP. Additionally, the number of sequences assigned to eukaryotes decreased from past to present, and across sediment cores, all five samples that did not contain any sequences assigned to eukaryotes fall in the most recent time period (<2 cal ka BP). We recorded low numbers of sequences assigned to codfishes in the five Melville Bay 26G samples from the lower, laminated part of the sediment core (>213 cm; Supplementary Fig. 11).

### Melville Bay
Using hybridization capture, we detected a total of ten different marine mammal species in the Melville Bay 26G record (Fig. 2e and S14). No DNA was assigned to marine mammals in the six samples from the lower, laminated part of the sediment core (>213 cm). The earliest detected marine mammal DNA was assigned to ringed seals (≥11.6–11 cal ka BP). Between 11.4–7.3 cal ka BP, we detected at least three different species in samples with marine mammal DNA. In a sample with estimated age 11.4 ± 1.2 cal ka BP, we recorded the highest

number of sequences assigned to one species (670 sequences assigned to harp seal). In a sample with estimated age 7.3 ± 0.6 cal ka BP, we recorded DNA from eight different marine mammals (Fig. 2e). After 7 cal ka BP, we detected fewer species, especially 3–0 cal ka BP. Between 8–7 cal ka BP, we recorded fin whale DNA (north of the species' current distribution range) and DNA of harp seal, beluga whale, minke whale, and orca; all of which only occur seasonally in the region today (Fig. 2e; Supplementary Data 5).

The fossil remains of foraminifera, single-celled protists that live in either the water column or at the seafloor, can be used to reconstruct paleoceanographic conditions[49–53]. For our analysis of Melville Bay 26G, we grouped the following foraminiferal species to indicate influx of Atlantic-derived water masses indicative of relatively warmer and saltier sub-surface waters associated with the West Greenland Current[36,61]: the calcareous species *Cassidulina neoteretis*, *Cassidulina reniforme*, *Islandiella norcrossi*, and *Pullenia osloensis*, as well as the agglutinated species *Adercotryma glomerata*, *Lagenammina diffugiformis*, *Reophax catella*, *Reophax pilulifer*, and *Reophax fusiformis*. We recorded their absence or near-absence in the oldest samples (>11 cal ka BP), high counts from 10–5 cal ka BP and low counts in the Late Holocene (<5 cal ka BP) (Fig. 2c, and Supplementary Data 6).

### Hall Basin and Lincoln Sea
The earliest detected marine mammal DNA in the Hall Basin 24PC record was assigned to ringed seals (7.2 ± 0.1 cal ka BP) (Fig. 3e and S14). We detected narwhal and ringed seal using hybridization capture in all Hall Basin 24PC samples <5 cal ka BP, albeit with decreasing sequence abundance over time (Fig. 3e). Between 5 cal ka BP and 3 cal ka BP, we also recorded harp and hooded seal DNA, two species which have only recently been recorded this far north[62]. In addition, we detected beluga whale DNA at 2.4 ± 0.0 cal ka BP, which has never been recorded this far north in present-day and historic surveys (Supplementary Data 5). The first detection of marine mammals (ringed seal) in the Lincoln Sea 12-GC record using hybridization capture is 8.4 ± 0.1 cal ka BP, and ringed seal DNA is recorded in all but one younger sample (Fig. 3h). Narwhal DNA was first identified at 7.9 ± 0.1 cal ka BP, and was detected in all but one younger sample, again with decreasing sequence abundance over time. We also detected DNA assigned to hooded seals in three samples (7.8 ± 0.1, 7.1 ± 0.1 & 1.8 ± 0.1 cal ka BP), albeit only with a low number of sequences and low relative sequence abundance.

### North-East Greenland
The hybridization capture data from the North-East Greenland 73G record revealed DNA of four endemic Arctic marine mammals (Fig. 4e and S14). We also retrieved DNA of harp seal (Fig. 4), which seasonally migrates northwards along the East Greenland coast (Supplementary Data 5). Most samples of this core are characterized by low numbers of DNA sequences (3–10 sequences per species per sample). For example, ringed seal DNA was detected throughout the core (13 out of 38 samples) but samples always contained <50 sequences. In a sample with estimated age 7.9 ± 0.2 cal ka BP, narwhal DNA was first retrieved, but only samples <4.5 cal ka BP showed a continuous genetic signal of narwhals. DNA from harp seals was detected in samples >3 cal ka BP. The highest number of sequences (145 sequences) was assigned to bearded seals in the youngest sample (0.3 ± 0.1 cal ka BP).

### DNA detections in the context of changing paleoenvironmental conditions
The statistical analysis of our DNA detections in conjunction with reconstructed air temperatures, sea-ice biomarker IP$_{25}$, primary productivity biomarkers brassicasterol and dinosterol, and foraminifera assemblages, revealed correlations between DNA detections and interpolated paleoenvironmental measurements, and identified air temperature and sea-ice algae productivity (IP$_{25}$) as significant drivers of community change (Supplementary Fig. 15 and S16; Supplementary

Data 7). Based on the pairwise Spearman's rank correlation analysis, we found strong positive correlations between e.g., codfishes and pinnipeds ($r_s$ = 0.8; $p < 0.01$), codfishes and ringed seals ($r_s$ = 0.7; $p < 0.01$), codfishes and interpolated air temperatures ($r_s$ = 0.6; $p < 0.01$), and priapulids (Priapulidae) and bowhead whales ($r_s$ = 0.7; $p < 0.01$) (Supplementary Fig. 15; Supplementary Data 7).

The redundancy analysis (RDA) of DNA data revealed air temperature and sea-ice algae productivity ($IP_{25}$) to be significant explanatory variables (air temperature: F = 2.52; Pr( >F) = 0.04; $IP_{25}$: F = 3.50; Pr( >F) = 0.02) albeit explaining only ~2% (air temperature) and ~3% ($IP_{25}$) of the observed variance (Supplementary Fig. 17).

## Discussion
We recovered a unique time series of marine mammal species occurrence covering the past ~12,000 years using a combination of shallow shotgun sequencing and mitochondrial hybridization capture of sedimentary ancient DNA from four coastal marine sediment cores around Northern Greenland.

### Ecosystem-wide changes in Melville Bay during the Holocene
The analysis of the Melville Bay 26G core, retrieved from the continental shelf of North-West Greenland (Fig. 1), revealed the paleoceanographic conditions and marine mammal communities throughout the entire Holocene, following the last deglaciation (Fig. 2).

The six oldest samples in the laminated section of this core ( > 213 cm depth in core) did not contain marine mammal DNA, nor did we detect foraminifera associated with Atlantic-derived water masses, but we detected low numbers of DNA assigned to codfishes and priapulids in the shotgun sequencing data (Supplementary Fig. 11d). During the Last Glacial Maximum (~26.5 to ~19 cal ka BP), the Greenland Ice Sheet extended to the edge of the continental shelf in Melville Bay, and only retreated from the outer coast by 11.6 ± 0.3 cal ka BP, when the marine-based ice sheet collapsed in Melville Bay[63]. In marine sediment cores, retreating ice-sheet deposits can be identified by a transition of laminated sediments containing no or low amounts of foraminifera to homogeneous sediments with dropstones and foraminifera[58]. The six samples retrieved from the laminated section of the core >213 cm thus originate from retreating ice-sheet deposits characterized by high sedimentation rates and therefore likely represent a short time period at or just prior to 11.6 ka BP.

The first marine mammal DNA we detected was from ringed seal at ≥11.6–11.0 cal ka BP. This finding pre-dates the earliest Holocene fossil evidence (K-4687; 8680 ± 120 BP[64]; 8.9 ± 0.4 cal ka BP; Supplementary Data 8) of this species in Greenland—and indeed of any marine mammal species—by ~2 ka. The same sample revealed DNA of codfishes, diatoms, and pinnipeds in the shotgun sequencing data (Supplementary Fig. 11d). For our DNA findings of ringed seals and codfishes, we identified a positive correlation, not just for Melville Bay 26G, but across samples from all four cores (Supplementary Fig. 15 and Supplementary Data 7). The analysis of Melville Bay 26G for foraminifera revealed near-absence before 10.5 cal ka BP (Data sheet 3, Fig. 2c), albeit an increase in the presence of foraminifera had been recorded in marine sediments of northernmost Baffin Bay at ~12.3 cal ka BP[50,65]. Ringed seals are physiologically adapted to life in the pack ice and are at present found in the Arctic year-round[66]. Our finding supports the hypothesis that ringed seals were the first marine mammals to move north after the end of the Last Glacial Maximum[38].

Between 11.4–7.3 cal ka BP, we detected DNA of marine mammals that are at present found in the region year-round (e.g., narwhal), only seasonally (e.g., harp seal), or not at all (fin whale) (Fig. 2e, Supplementary Data 5), potentially reflecting a non-analogous community structure in the Early Holocene as compared to recent and historic surveys[14,15,67,68]. This period also had the highest diversity of detected eukaryotic families and the highest abundance of foraminifera

associated with relatively warmer waters of Atlantic origin, which indicates increased intensity of the West Greenland Current (Fig. 2).

The detection of narwhal DNA at 11.4 ± 1.2 cal ka BP predates by over four millennia the earliest Holocene fossil record for narwhals from the region: a fossil from northern Ellesmere Island, Canada (82° N) (TO476; 6,830 ± 50 BP[69]; 6.9 ± 0.3 cal ka BP; Supplementary Data 8) and a younger fossil from the northern tip of Greenland (83.65° N) (AAR11912; 5,845 ± 60 BP[35]; 5.9 ± 0.3 cal ka BP; Supplementary Data 8). Similarly, the detection of bowhead whale DNA at 10.7 ± 1.0 cal ka BP predates the earliest fossil record of this species in Melville Bay (LuS-6443; 8,525 ± 50 BP[70]; 9.0 ± 0.3 cal ka BP; Supplementary Data 8) by at least 400 years. There is a relative scarcity of bowhead whales in the Greenland fossil record, but their fossil occurrence in the Canadian Arctic Archipelago documents their presence at similar latitudes (~73° N) as Melville Bay (~75° N) as early as 10.5 ka BP[71,72].

Temperate cetaceans, including minke whales and orcas, are known to migrate northward along the west coast of Greenland during the summer months[17], albeit Melville Bay is at the northern limit of their present-day distribution range (Supplementary Data 5). Fin whales, which we detected at 7.8 ± 0.7 cal ka BP, have not previously been recorded along the west coast of Greenland further north than 71° N in present-day and historic surveys[16,73,74]. In addition to temperate marine mammals, we also detected DNA from the low-arctic hooded seal (at 11.4 ± 1.2 and 4 ± 0.4 cal ka BP), harp seal (11.4–0.9 cal ka BP), and beluga whale (7.9–3 cal ka BP). Both hooded and harp seals migrate northward along the west coast of Greenland during summer, but in contrast to harp seals, which forage in shallower waters[15,75], hooded seals generally stay further offshore[14]. For beluga whales, Melville Bay is known as a migration corridor between their summer distribution hotspot in the Canadian Arctic and their wintering grounds further south along the west coast of Greenland[68].

The detection of temperate and low-arctic marine mammal species suggests that during the Early-to-Mid Holocene, they were present in the area at densities that are detectable in marine sediments. We hypothesize that this expansion or northward shift in their distribution ranges was associated with environmental changes. Specifically, the redundancy analysis (RDA) of DNA detections and reconstructed paleoenvironmental proxy measurements revealed air temperature and sea-ice algae productivity ($IP_{25}$) to be significant drivers of community change (Supplementary Fig. 16); the biomarker $IP_{25}$ is a lipid produced by a group of Arctic sea-ice associated diatoms, and its concentration in sediments can provide information on the extent and nature of past sea ice cover[76].

Furthermore, our analysis of foraminiferal assemblages from Melville Bay 26G revealed high foraminiferal counts, indicating increased intensity of the West Greenland Current 10–7 cal ka BP (Fig. 2c), which is in line with previous paleoceanographic reconstructions from northernmost Baffin Bay[50,65,77]. The biomarkers brassicasterol and dinosterol (both reflecting primary productivity), and $IP_{25}$ were also relatively high in Melville Bay between 10.5–6 cal ka BP, indicating a phase of high primary productivity and stable ice-edge conditions[76,78]. The West Greenland Current (Fig. 1a) transports relatively warmer waters of Atlantic origin northwards along the west coast of Greenland[79] and its prevalence has been shown to affect the size and abundance of marine organisms at the base of the food web[80].

Our detections in the Early-to-Mid Holocene of temperate species such as orcas, fin whales, and minke whales at 75° N may reflect a northward expansion relative to their present-day distribution, similar to the occurrence of boreal mollusc species along the west coast of Greenland (9.2 - 5.6 cal ka BP), which has been interpreted as a consequence of higher sea-surface temperatures associated with an increased intensity of the West Greenland Current[81]. To date, the mechanisms controlling increased northward heat transport are not yet fully understood, but certain atmospheric conditions can facilitate increased intensity of the West Greenland Current[82].

Overall, our DNA assemblage from Melville Bay 26G contained both Arctic and temperate species, which may indicate the coexistence of species in non-analogous communities. However, the age uncertainty of our individual sediment samples from this core span at least several hundred years (ranging from 0.4–1.6 ka) and therefore we cannot rule out that replacement of species may have occurred at a finer temporal scale (Fig. 2e).

In Disko Bay (69° N), an increased prevalence of temperate cetaceans such as humpback whales has been observed during the summer months since the 18th century[83]. Humpback whales, which are found at lower latitudes during most of the year, have been observed in the shallower waters of Disko Bay later in the summer, whereas bowhead whales migrate to Disko Bay earlier and spend more time in deeper waters. The co-occurrence rather than replacement of species may be facilitated by bowhead whales feeding on copepods and amphipods, while humpback whales target krill and fish[83,84]. However, further south, off the coast of South-East Greenland, the increased presence of temperate marine mammal species correlates with a decrease in Arctic species[6].

Following this period of higher species diversity, marine mammal detections were more sporadic in the Mid-to-Late Holocene (7.2 - 3 cal ka BP), although we still detected species occurring seasonally in the region today (Fig. 2e). In the same period, we detected less eukaryotic families (Supplementary Fig. 11d), and the abundance of foraminifera indicative of increased intensity of the West Greenland Current decreased (Fig. 2c). The timing matches the disappearance of boreal mollusc species in western Baffin Bay after 5.6 cal ka BP[81] and eastern Baffin Island after 3 cal ka BP[72], both attributed to a decline in sea-surface temperature. Similarly, sea-surface temperature reconstructions based on diatom and dinocyst assemblages of a sediment core from northernmost Baffin Bay showed a transition from fluctuating but higher temperatures >4 cal ka BP, to lower temperatures <4 cal ka BP[77].

After 3.2 cal ka BP, marine mammal detections were rare and we only detected DNA of species that occur seasonally in the region today in one sample (harp seal at 0.9 cal ka BP). In this period, our Melville Bay 26G foraminifera data suggests low influx of Atlantic-derived water masses, potentially reflecting a phase of a weaker West Greenland Current, as has been suggested based on a paleo-oceanographic reconstruction using another marine sediment core from southern Melville Bay[49]. Furthermore, data from several other marine sediment cores indicate a phase of increasing sea ice concentrations and decreasing phytoplankton production during the Late Holocene in northern Baffin Bay[52,65,85] and Disko Bay[36]. A shift from low-arctic species such as harp seal and harbor porpoise >3 cal ka BP to Arctic endemic marine mammals such as ringed seals and bowhead whales <3 cal ka BP has also been described based on the analysis of archeological middens[86,87]. The change in the consumption of marine mammals by Paleo-Inuit communities along the west coast of Greenland may reflect the species that were most prevalent in the environment at a certain time period.

Only two samples had estimated ages <2.2 cal ka BP, and thus, we have limited temporal resolution for the most recent past for the region. However, one sample (2.2 ± 0.7 cal ka BP) with DNA of narwhal, bearded seal, and ringed seal falls within the Roman Warm Period, which is a period of relatively higher temperatures in the context of the Late Holocene ~2.2 –1.3 ka BP[88], during which marine sediment cores in Baffin Bay have recorded increased intensity of the West Greenland Current[36,50]. In contrast, the youngest sample (0.2 ± 0.3 cal ka BP) contained only ringed seal DNA and fell within the Little Ice Age–the most recent cold period ca. 0.6–0.1 ka BP[88]–where low intensity of the West Greenland Current has been recorded[36,50]. The absence of marine mammals in periods of colder climates and lower intensity of the West Greenland Current could point to a southward shift in distribution, as has been identified for bowhead whales during the Little Ice Age based on fossil remains[89].

## The opening of the Nares Strait

Our two northernmost cores, Lincoln Sea 12-GC (82.58° N) and Hall Basin 24PC (81.62° N) (Fig. 1a), provide unique insights into the Holocene history of marine mammal populations along the northernmost coast of Greenland. Today, the region is regarded as part of the 'Last Ice Area', where the persistence of multiyear sea ice may provide a refuge for Arctic species under global warming[90]. Towards the end of the Pleistocene (19 - 13 ka BP), the narrow connection between northernmost Baffin Bay and the Lincoln Sea known as the Nares Strait was covered by shelf-based ice extending between the Ellesmere Island and Greenland ice sheets[34]. The retreat of shelf-based ice marks the opening of the Nares Strait ~9 ka BP, although sea ice is still pronounced throughout most of the year today[51,91].

The Lincoln Sea 12-GC core, retrieved North-East of Hall Basin 24PC, extends back ~12 ka BP, whereas Hall Basin 24PC only covers the past ~9.6 ka BP (Fig. 1c). We did not detect any marine mammal DNA in Lincoln Sea 12-GC >8.4 cal ka BP (Figs. 3e, h and S14); the Early Holocene has been characterized as a period of seasonal sea ice and relatively higher primary productivity in the southern Lincoln Sea[56,92]. Even if the previous detection of the sea-ice biomarker $IP_{25}$ and primary productivity biomarker brassicasterol in the same core corresponding to the Early Holocene indicate suitable sea-surface conditions[56], marine mammals may not have been able to access the habitat as long as the Nares Strait was covered by glacial ice. Our first detection of marine mammal DNA (ringed seal; 8.4 ± 0.1 cal ka BP) supports this, occurring shortly after the opening of the Nares Strait ~9 cal ka BP[51,91]. At the same time, the detection of ringed seal DNA is in line with our findings for Melville Bay 26G and provides further evidence that ringed seals are the first marine mammals to move to newly accessible habitats (Supplementary Fig. 14)[38].

We provide evidence for the occurrence of narwhals in the Lincoln Sea by 7.9 ± 0.1 cal ka BP; we detected narwhal DNA in all samples <7.9 cal ka BP from Lincoln Sea 12-GC (except one sample with estimated age 5.3 ± 0.2 cal ka BP), and in five out of ten Hall Basin 24PC samples, indicating the presence of the species in the area at a density that is detectable in the cores (Fig. 3e, h). Our findings suggest colonization of the newly accessible habitat by populations migrating north from Baffin Bay through Nares Strait occurred within a few hundred years of the strait opening. Although single narwhal observations have documented the species´ presence in the Nares Strait[93–95], their northernmost recognized management unit (Smith Sound) lies further south ( ~78° N) and only partly includes Nares Strait[68]. As our understanding of past and present distributions of narwhals improves, the consequences for current management units and marine protected areas will need to be considered, especially in the light of expected future northward distribution shifts[8,96].

In the Hall Basin 24PC core (81.62° N), our DNA detections indicate the establishment of ringed seals after 7.2 ± 0.0 cal ka BP (Fig. 3e), and a later establishment of narwhals and hooded seals than for Lincoln Sea 12-GC, albeit the period 8.6– 5 cal ka BP was only covered by two samples (7.2 & 8.6 cal ka BP, Fig. 3f). A possible explanation may be the earlier retreat of the Ryder Glacier from the mouth of Sherard Osborn Fjord (estimated at >10.7 ± 0.4 ka cal BP)[32] relative to the retreat of the Petermann Glacier from the Petermann fjord mouth (estimated at 7.5 cal ka BP)[59]. Although the Lincoln Sea 12-GC core (in proximity to the Ryder Glacier) and Hall Basin 24PC core (in proximity to the Petermann Glacier) were collected outside the actual fjords, the dynamics of the two glaciers during the Holocene have had a strong impact on the marine ecosystems and depositional environment, as seen e.g., by the dramatic increase in the abundance of foraminifera after 8.5 cal ka BP in the Hall Basin marine sediment core OD1507-18GC[58].

In the Mid-to-Late Holocene (5–2 cal ka BP), Hall Basin 24PC recorded the presence of harp seal, hooded seal, and beluga whale: three migratory species to the northernmost part of Baffin Bay, of

which only harp and hooded seal have been observed north of 80° N in present-day and historic surveys (Supplementary Data 5)[62,68]. Albeit based on a single sample, the detection of beluga whale DNA in Hall Basin 24PC at 2.4 ± 0.0 cal ka BP coincides with the timing of higher air temperatures (+0.7 °C as compared to the pre-industrial average[29]), suggesting a northward distribution shift of the species during past warmer climates. This matches habitat suitability models, which predict a northward shift of suitable habitat from areas where beluga whales occur in summer today (~75° N) into parts of Nares Strait (~79° N), assuming warming <2 °C as compared to the pre-industrial average[8].

In the Lincoln Sea 12-GC core, which is further north (82.58° N) than Hall Basin 24PC, all samples with hooded seal DNA were >0.9 ± 0.3 cal ka BP, prior to the most recent re-advance of the Ryder glacier and the re-growth of its ice tongue to the outer sill of Sherard Osborn fjord at <0.9 ka BP[32]. Generally cooler conditions and more severe ice conditions in and around the Lincoln Sea may have negatively impacted habitat suitability for hooded seals.

### Sporadic marine mammal detections in North-East Greenland over the past 10 ka

The marine ecosystem of the North-East Greenland shelf is shaped by the East Greenland Current (Fig. 1a), which transports cold water masses and >90% of the sea ice exported from the Central Arctic Ocean into the North Atlantic[97], and a deflected branch of the West Spitsbergen Current, which transports relatively warm and saline water masses from the North Atlantic[98].

In sediment core North-East Greenland 73G (79.07° N), we sporadically detected DNA of five marine mammals over the past 10 ka, with samples generally containing low numbers of sequences (Fig. 4e and S14). Between 10–4 cal ka BP, nine of 21 samples contained marine mammal DNA, of which only four contained DNA of more than one species. In the same period, the increase of sea-ice biomarker $IP_{25}$ (Fig. 4a) after 8 cal ka BP indicates a shift from open water to higher sea ice cover conditions[99], while higher concentrations of the biomarker brassicasterol (Fig. 4b) and higher abundances of Atlantic water foraminifera (Fig. 4c) between 10–9 cal ka BP and 8–5 cal ka BP indicate periods of higher primary productivity and elevated Atlantic water influence[57,99]. Lower concentrations of the primary productivity biomarkers brassicasterol and dinosterol as compared to sediment cores from Melville Bay (Fig. 2a–c) highlight differences in the oceanographic conditions between the West and East Greenland shelf. The West Greenland shelf is influenced by chilled Atlantic water import through the West Greenland Current, while the East Greenland shelf is predominantly influenced by cold water and sea ice import through the East Greenland Current[97] (Fig. 1a). The near-absence of marine mammal DNA over the past 10 ka (Fig. 4e) indicates lower habitat suitability of the East Greenland shelf as compared to the West Greenland shelf, despite the observed variability in the paleoceanographic reconstruction (Fig. 4a–c).

We only detected bowhead whale DNA in one sample 5.7 ± 0.2 cal ka BP, although the current presence of the species in this region has repeatedly been reported[100,101]. It has been hypothesized that bowhead whales were present in North-East Greenland in the Early Holocene[38], based on their Early Holocene presence in Svalbard[102], where the timing of the last deglaciation (and thus the availability of suitable habitat) was similar to North-East Greenland[103,104]. However, few bowhead whale fossils have to date been found in east Greenland; one Early Holocene record further south (72.13° N; Lu-1095; 8,580 ± 85 BP[105]; 9.0 ± 0.3; Supplementary Data 8), one Mid Holocene at the same latitude as our core (79.57° N; K-6891; 5,190 ± 70 BP[106]; 5.4 ± 0.3 cal ka BP; Supplementary Data 8) and one Mid Holocene record further north (82° N; 6,050 ± 105 BP[39]; 6.3 ± 0.3 cal ka BP; Supplementary Data 8).

A key difference between marine ecosystems around Svalbard and North-East Greenland is the interplay of currents transporting cold

and warm water masses, leading to sea surface temperatures ~5 °C lower on the North-East Greenland shelf, as compared to the Svalbard shelf[107]. Our low genetic detection of bowhead whales in marine sediments and rare fossil discoveries in the region suggest a limited past presence of the species along the east coast of Greenland.

After 4.5 ka BP, we detected narwhal DNA in ten of 20 samples, and ringed seal DNA in nine of 20 samples (Fig. 4e), while we did not identify DNA assigned to codfishes or harp seal in any of the North-East Greenland 73G samples with estimated ages <3 cal ka BP. This contrasting pattern may indicate that lower air temperatures in the Late Holocene (Fig. 1b) and higher sea ice cover in the region (Fig. 4a) caused a north-ward distribution shift of narwhals and ringed seals and a south-ward distribution shift of codfishes and harp seals.

The youngest sample (0.3 ± 0.1 cal ka BP) contained the highest number of sequences of any marine mammal (bearded seal) in this core. Both sea-ice algae productivity and phytoplankton productivity increase slightly after 1 ka (Fig. 4a–b), pointing towards a recent shift in the environmental conditions that has been interpreted as reflecting the formation of the North-East Greenland polynya[99]. Polynyas are annually recurring ice-free areas in high latitudes characterized by high primary productivity that can support high abundances of Arctic species communities, including marine mammals[85,108]. In contrast to the North Water polynya (located in northernmost Baffin Bay), the history of the North-East Greenland polynya has so far received less attention[99]. Our findings of sporadic rather than continuous detection of marine mammal DNA in the region encompassing the North-East Greenland polynya potentially reflects the instability of this marine ecosystem and vulnerability to climatic changes as has been inferred for the North Water polynya[85].

Our findings provide insights into 12,000 years of marine mammal occurrence around Northern Greenland. We report the increased prevalence of low-arctic and temperate species during the Early-to-Mid Holocene at Melville Bay 26G and North-East Greenland 73G, a period with regional high air temperatures, and the presence of foraminifera associated with increased influence of Atlantic-derived warmer water masses, and higher primary productivity. Based on data from Lincoln Sea 12-GC and Hall Basin 24PC, we reconstruct the timing of the first establishment of marine mammal populations after the opening of the Nares Strait ~9 kya. We detect the earlier occurrence – in some cases by several thousands of years– of several marine mammal species in Northern Greenland relative to their fossil chronology. During the Late Holocene, we detect fewer marine mammal species at all four sites, likely reflecting the decreased influence of Atlantic-derived warmer water masses and lower primary productivity during the neoglaciation. Our study demonstrates the potential of sedimentary ancient DNA for providing long-term baseline data of marine mammal occurrences, and for improving our understanding of the effects of past environmental changes on species distributions and community composition.

## Methods
### Permits and permissions

In compliance with the Biological Diversity Convention, the Nagoya Protocol and the Greenland Parliament Act No. 3 of 3$^{rd}$ June 2016, we obtained Prior Informed Consent licenses for the collection and use of genetic resources in West Greenland (non-exclusive licence no. G21-041 and G19-009) from the Ministry of Foreign Affairs, Business, Trade and Climate of Greenland. This includes an export permit for the genetic resources. For Northeast Greenland, we obtained permits to enter the National Park and conduct biological investigations from the Ministry of Nature and Environment of Greenland and the Ministry of Industry, Labor and Trade, respectively (permits no. C-17-61 and C-19-50) as well as permits from the Mineral Licence and Safety Authority for geological investigations (permit no: VU-00125) and for the collection and export of sediments (permit no: 129/2017). Further, a research survey permit to enter the Northeast Greenland National Park

was provided by the Greenlandic/Danish authorities (Permits C-19-50 and JTHAV 2019-14122) and for Canadian waters, surveys operated under permit IGR-940.

## Sediment cores

We analyzed four marine sediment cores in the present study, which were collected off the coasts of Northern Greenland (Fig. 1a). All cores were cut into 1 m sections onboard the research vessels upon retrieval, and stored at 4 °C until subsampling for DNA was carried out. The four cores were collected during three expeditions:

Gravity core LK21-IC-st26-GC1 (75.319° N 61.910° W, 912 m water depth, 320 cm sediment recovery; hereafter Melville Bay 26G) was retrieved from the North-West Greenland shelf during the ICAROS Expedition onboard *HDMS Lauge Koch* in 2021. A chronology was constructed using a combination of $^{210}$Pb and $^{14}$C dating (Supplementary Fig. 8). The upper 10 cm were tested for $^{210}$Pb activity (Supplementary Data 3) using a Canberra ultralow-background Ge-detector using the Constant Rate of Supply (CRS) model[109]. The radiocarbon analysis was carried out using an accelerator mass spectrometer (AMS) mini carbon dating system (MICADAS) with gas targets[110] on three mixed planktonic and three mixed benthic foraminifera samples (Supplementary Data 2). The age-depth model was modeled in R[111] using a Bayesian accumulation model code (BACON)[112] calibrated with Marine20[113] and a local reservoir correction (ΔR) of -49 ± 59 years[114].

Piston core Ryder19-24-PC1 (81.622° N 62.296° W, 520 m water depth, 525 cm sediment recovery; hereafter Hall Basin 24PC) was retrieved from the North Greenland shelf during the Ryder 2019 Expedition onboard R/V *Oden*. It was taken at the same station (450 m away) as OD1507-18GC, which had been collected by R/V *Oden* during the Petermann 2015 expedition. The motivation for re-coring the site was two-fold; 1) to recover the basal diamict at this site, and 2) to collect fresh material for organic geochemical and sedimentary ancient DNA work. The radiocarbon dates from OD1507-8GC were migrated onto the depth scale of Hall Basin 24PC, after a common depth scale was constructed through the correlation of bulk density, magnetic susceptibility, and XRF-scanning data (Supplementary Fig. 9, Supplementary Data 4). The $^{14}$C ages of OD1507-8GC[58] were calibrated using the Marine20 calibration curve[113]. We applied reservoir corrections of 270 ± 80 years (planktic) and 510 ± 90 years (benthic) consistent with previous chronologies in northern Nares Strait[58,59], but subtracted 150 years from these to account for the differences between the Marine13 and Marine20 calibrations[113] (Supplementary Data 4). The top of the basal diamict was assigned an estimated age of 8910 ± 100 years BP, corresponding to the inferred ice margin location just north of the core site in the Polaris Foreland region[60]. This age estimate was inferred by England[60] based on a regional compilation of radiocarbon dates of mollusc shells, and further validated by Jakobsson et al.[59] through bathymetric mapping and sub-bottom profiling of Hall Basin. We note that the published ice limit was assigned an age of 8,500 BP[60], but this was after a reservoir correction of 410 years had been subtracted. The final age model (Supplementary Fig. 10) was generated using *OxCal*[115]. It displays a marked reduction in sedimentation rates after the transition from the laminated to the bioturbated mud units (BI and BM) consistent with the reduction in sedimentation rates through the Mid-to-Late Holocene in marine records from this region[58]. The age for the base of the bioturbated muds is consistent with other records from the Hall Basin[58].

Gravity core Ryder19-12-GC1 (82.578° N 52.528° W, 867 m water depth, 273.5 cm sediment recovery; hereafter Lincoln Sea 12-GC) was also retrieved during the Ryder 2019 Expedition and an age-depth model (Supplementary Fig. 6) and a paleoceanographic reconstruction based on sterols has been published[56].

Gravity core DA17-NG-ST07-073G (79.068° N 11.903° W, 385 m water depth, 410 cm sediment recovery; hereafter North-East Greenland 73G) was retrieved from the North-East Greenland shelf during the NorthGreen Expedition onboard R/V *Dana* in 2017. An age-depth model (Supplementary Fig. 7) and a paleoceanographic reconstruction based on foraminiferal assemblages has been published[57].

## Subsampling of cores for sedaDNA analysis

The subsampling of Melville Bay 26G and North-East Greenland 73G was carried out at the Globe Institute, University of Copenhagen, whereas subsampling of Hall Basin 24PC and Lincoln Sea 12-GC was carried out at the Center for Paleogenetics, University of Stockholm. In both events, a designated clean sub-sampling laboratory was used that is physically isolated from the molecular biology laboratories. All working surfaces and equipment used during the subsampling were soaked in bleach and subsequently cleaned with ethanol. All people involved in the subsampling process wore appropriate protective clothing, including lab coveralls, two layers of gloves, surgical face masks, and sleeves. We regularly changed gloves and used 5% bleach followed by ethanol to avoid contamination. Initially, each core section was split into one working half and one archive half. We carefully removed a thin layer of sediment (~0.2 cm) using sterile plastic cards first and another thin layer using sterile single-use scalpels just before the actual subsampling of the working half was performed. We subsampled using sterile 3 mL plastic syringes, and samples were immediately frozen to avoid further DNA degradation.

## Extraction and library preparation

All pre-PCR laboratory work was performed at the ancient DNA clean lab facilities at the Globe Institute, University of Copenhagen, where strict precautions are taken to avoid contamination[116,117]. A subset of samples was processed using the semi-automated ancient environmental DNA pipeline operated by the GeoGenetics Sequencing Core facility at the Globe Institute. For these samples, 0.35 g of sediment were subsampled and extracted using the Qiagen® MagAttract® Power Soil Pro kit, with modifications[118]. After extractions, DNA concentrations were quantified by qPCR and a fixed number of cycles for the library build was determined (17 cycles). The libraries were prepared following the double-stranded protocol[119] and unique 10-base pair motifs were used for double-indexing. The amplified libraries were purified and size selected with magnetic beads (MagBio Genomics), targeting 60-600 base pairs (bp; with a 1.6x ratio) and fragment lengths and concentrations were determined using a Fragment Analyzer (Agilent Technologies).

The remaining samples were processed manually using the Qiagen DNeasy® PowerSoil® Pro Kit following the product protocol with minor modifications: an input weight of 0.5–1.0 g sediment was used. We added 25 μL of DTT (Thermo Scientific™; 1 M) and 25 μL of protK (Sigma-Aldrich®; 2 mg mL$^{-1}$) to the bead tube containing the beads and sediment sample. The vortex step was performed using a FastPrep-24™ 5 G with 2 × 20 s (4 m s$^{-1}$). Samples were incubated for ~24 h at 56°C. In the last step of the protocol, 35 μL of elution buffer was added followed by a 5-min incubation. This step was performed twice to yield a total extract volume of 70 μL.

Single stranded libraries were prepared from 1 - 14 ng of DNA (as determined by Qubit dsDNA HS assay; Thermo Scientific™) following the Santa Cruz Reaction protocol[120]. For the indexing, two unique 6-bp motifs were used for each library, to minimize the risk of cross-contamination during pooling for sequencing. Blanks were included in both workflows to investigate potential contamination. For shotgun sequencing, indexed libraries were pooled equimolarly (except for blanks which were included with 10% molarity) and sequenced by NovoGene UK on an Illumina NovaSeq 6000 platform.

## Hybridization capture

Shotgun sequencing revealed a low number of mammalian sequences, thus we applied hybridization capture to enrich libraries specifically for mitochondrial DNA derived from marine mammals

occurring in Greenland seasonally, as well as year-round (Supplementary Data 5 & 9). In preparation of the panel design, which was conducted in collaboration with Daicel ArborBiosciences myBaits®, the first 80 nucleotides of each mitogenome were added to the end of each mitogenome sequence of interest. Next, runs of ambiguous nucleotides were changed to thymine nucleotides. Finally, to account for the circularity of mitochondrial genomes, a 4x tiling with 80 nt baits was achieved by starting a new bait every 20 nt, resulting in 10,254 unique baits.

The hybridization capture panel used in this study was originally designed for the retrieval of mitochondrial DNA from macrofossils, and therefore includes several species for which we did not expect to find DNA in marine sediments (e.g., polar bears, *Ursus maritimus*, which mostly live on sea ice and occur in low densities). At the same time, the panel did not include species occurring along the coasts of Greenland in large numbers (e.g., harp seals). However, by lowering the hybridization temperature to 55 °C, we could facilitate the retrieval of DNA from closely related species with up to 25% genetic divergence (myBaits® User Manual (v 5.02)), while also accommodating for the expected DNA damage of marine sedimentary ancient DNA[47].

We followed the "High Sensitivity" protocol of the myBaits® User Manual (v 5.02) with the following additional modifications: the input mass of each indexed library into each capture reaction was 154 ng. Generally, higher inputs are possible (up to 12 µg per enrichment reaction) but we standardized the input mass to account for libraries with lower DNA concentrations across our sample set. The input volume of baits and water in the hybridization mix was adjusted to 1.1 µL and 4.4 µL, respectively. One round of capture was performed.

We repeated the lab procedure (DNA extraction, library preparation, hybridization capture, and sequencing) for 22 samples to validate the reproducibility of marine mammal detections even with low numbers of DNA sequencing reads. After confirming high overlap of species detections across samples (Supplementary Data 10), we merged the sequencing data of samples deriving from the same depth in the sediment and core after collapsing sequencing pairs and removing sequencing adapters.

## Bioinformatic Analysis

**Shotgun sequencing.** Raw sequencing data were processed using *leeHom*[121] to trim adapters and merge sequencing reads. Merged sequences were complexity-filtered, sequences shorter than 30 bp were discarded and exact duplicates and homopolymers were removed using *sga*[122]. For taxonomic assignment of the shotgun sequencing data, a reference index was constructed from the non-redundant NCBI nucleotide database (downloaded on 1st December 2023), the full NCBI RefSeq database (release 213), and a compilation of Arctic plant and animal genomes[45]. Filtered sequences were taxonomically assigned using *bowtie2*[123] and alignment files were compressed using *compressbam* of the *metaDMG*[124] suite.

The compressed files were parsed to *metaDMG*[124], which was used to filter alignments using a lowest-common-ancestor approach with a 95% sequence similarity threshold and to assess DNA damage. All downstream visualizations were performed in R[111]. The taxonomic assignments of the shotgun sequencing data were additionally filtered by a minimum of 10 sequences per taxon and for Fig. S7-S9, only taxa present in at least 2 samples across each core were visualized (for the full list of DNA detections, please refer to Supplementary Data 1). Furthermore, we excluded all taxonomic assignments to prokaryotes, thus only retaining detections of eukaryotes. We calculated relative sequence abundances for each taxon per sample by dividing the number of unique sequences by the number of quality-filtered read pairs, thus providing a representation of the sequencing effort and library complexity.

Using the *metaDMG* output files, we extracted nucleotide deamination rates, and calculated the mean value across the last three nucleotide bases of both 3' and 5' ends of each DNA sequence. For all eukaryotic sequences, we visualized these mean values to provide a representation of DNA damage across sample ages and sediment cores (Supplementary Fig. 1a-d).

**Mitochondrial capture.** The raw mitochondrial capture sequencing data were processed in the same manner as the shotgun sequencing data, with the following modifications: for taxonomic assignments, a reference database was constructed using all complete vertebrate mitochondrial genomes and the complete refseq database of mitochondrial sequences (downloaded from the NCBI databases on 17th Oct. 2023). To reduce the size of the resulting collection of reference sequences, we retained only one *Homo sapiens* mitochondrial genome in our database. After the taxonomic assignment, an additional duplicate removal step was included, where sequences with the same start and end position were removed using *samtools*[125].

We also performed an analysis of sequence similarity on mitochondrial genomes to establish a sequence similarity threshold appropriate for the observed genetic divergence within the target group of Arctic marine mammals (Supplementary Fig. 17). Using a low sequence similarity threshold (e.g., 95%) as part of the lowest-common-ancestor inference for a target group with moderate genetic divergence between species (interspecific variation) and low genetic divergence within the same species (intraspecific variation) may underestimate the number of detected species, as sequences are classified on a higher taxonomic level (e.g., genus or family). In the same scenario, a high sequence similarity threshold (e.g., 100%) may overestimate the number of detected species based on single nucleotide mismatches deriving from sequencing errors or DNA damage[126]. Specifically, we downloaded all publicly available mitochondrial genomes belonging to the subfamilies Phocinae, Balaenidae and Monodontidae (downloaded from the NCBI on 25th Mar. 2024).

After aligning and trimming the genomes, we generated thousands of sequence fragments of 50 bp length from each genome. We calculated pairwise sequence similarities between the simulated fragments of mitochondrial genomes within the same species and between species. Within species, we observed that for any pair of mitochondrial genomes, 95% of the 50'mers showed sequence similarities between 98-100% whereas only 5% showed sequence similarities below 98% (Supplementary Fig. 17). In contrast, between species of the same subfamily, we observed that for any pair of mitochondrial genomes, only 10% of the 50'mers showed sequence similarities above 98%, whereas 26% showed sequence similarities between 95–98%, and 64% showed sequence similarities below 95% (Supplementary Fig. 17). We thus argue that a 2% threshold is appropriate for our taxonomic target group, to accommodate for the observed high sequence similarity within species and moderate sequence similarity between species.

The resulting taxonomic assignments were filtered using a minimum of 3 unique sequences per taxa and for Figs. 2–4, only species-level marine mammal detections were visualized (for the full list of DNA detections, please refer to Supplementary Data 11). We derived mean nucleotide deamination rates for DNA sequences retrieved through hybridization capture as described for the metagenomic shotgun sequencing data, and included them in Supplementary Fig. 1. In addition, we binned the hybridization capture sequencing data by estimated age windows of 3000 years, and used *euka*[127] on the resulting files to analyze DNA deamination rates, fragment length distributions and coverage across the mitochondrial genome. We identified two taxa (Odontoceti & Phocidae) and visualized the respective DNA characteristics for all DNA sequences assigned to the taxonomic family Phocidae, as it yielded the highest number of sequences across binned samples and cores (Supplementary Fig. 2-5).

## Paleoenvironmental and paleoclimatic data for comparison

**Published records.** For comparison of our DNA data on marine mammal distributions, we compiled existing records on surface and bottom/subsurface water conditions at each study site. Paleoceanographic proxy data was retrieved from the literature for the same cores when available, or else from published records in proximity to the cores.

For Melville Bay 26G, measurements of sea-ice biomarker $IP_{25}$ (indicative of sea-ice algae productivity), brassica- and dinosterol (indicative of primary productivity) were compiled from a nearby marine sediment core (GeoB19927-3)[76].

For Hall Basin 24PC and Lincoln Sea 12-GC, a Holocene air temperature reconstruction from the Agassiz ice cap[29] and $IP_{25}$ as well as brassicasterol measurements from Lincoln Sea 12-GC[56] were compiled. In addition, trends in $\delta^{13}C$ values of the benthic foraminifera *Cassidulina neoteretis* and abundance of the benthic foraminifera *Nonionellina iridea*–indicative of the opening of Nares Strait–were compiled from HLY03-01-05GC[51].

For North-East Greenland 73G, measurements of $IP_{25}$, brassica- and dinosterol were retrieved from PS93/025-2[99]. Furthermore, we included the abundance of the two benthic foraminifera *Cassidulina neoteretis* and *Pullenia bulloides*–indicative of relatively warm, saline Atlantic water–previously published for North-East Greenland 73G[57].

**Analysis of foraminifera assemblage (Melville Bay 26G).** For Melville Bay 26G, no foraminiferal data existed and we carried out the analyzes for this study: ~four grams of sediment per sample depth (each sample representing 1 cm of the core) were washed and wet-sieved using a 63 μm sieve. Next, all constituents >63 μm were left in foraminifera storage solution (prepared using 300 ml of ethanol (96%), 700 ml of distilled water, and 1.5 grams of sodium carbonate) for ca. 30–40 minutes to dissolve any cohesive clay clusters that remained after sieving. Afterwards, all samples were washed once more with the foraminiferal storage solution before the foraminiferal assemblage was analyzed using an Olympus stereomicroscope. The following foraminiferal species were grouped to indicate influx of chilled Atlantic Water associated with the West Greenland Current[36,61]: the calcareous species *Cassidulina neoteretis*, *Cassidulina reniforme*, *Islandiella norcrossi* and *Pullenia osloensis*, as well as the agglutinated species *Adercotryma glomerata*, *Lagenammina difflugiformis*, *Reophax catella*, *Reophax pilulifer*, and *Reophax fusiformis*.

## Statistical analysis

We performed a statistical analysis on the association between DNA detections and available paleoclimatic data (Figs. 2a–c, 3a and 4a–c) using correlation analyzes and redundancy analysis (RDA) in R[111]. First, we approximated air temperature anomaly, $IP_{25}$, foraminifera abundance, brassica- & dinosterol measurements for each estimated sample age using linear interpolation, and standardized all interpolated values to have a mean of zero and a standard deviation of 1. We then calculated pairwise Spearman's rank correlation coefficients for all species-level detections (derived from hybridization capture), family-level detections (derived from shotgun sequencing) and interpolated paleoenvironmental and paleoclimatic measurements. The resulting correlation matrix was filtered for significant ($p < 0.05$) correlations (Supplementary Fig. 15; Supplementary Data 7).

In addition, we performed a Redundancy Analysis (RDA) on our data to estimate the contribution of each interpolated proxy measurement to explaining the diversity of the species-level detections based on hybridization capture, and family-level detections based on shotgun sequencing. The significance of the estimated contribution was evaluated using an ANOVA-like permutation test (Supplementary Fig. 16).

All published radiocarbon dates of marine mammal fossils were calibrated in R[111] using Marine20[113] and region-specific reservoir corrections (Supplementary Data 8).

## Reporting summary

Further information on research design is available in the Nature Portfolio Reporting Summary linked to this article.

## Data availability

The raw sequencing data analyzed in this study have been deposited at the Sequence Read Archive (SRA) under BioProject PRJNA1211513 and Supplementary Data 12 lists individual accession numbers of the respective sequencing libraries. Source Data for Figs. 2, 3, and 4 can be found in Supplementary Data 6 and 11. The palaeoenvironmental and palaeoclimatic data presented in Figs. 1–4 and Supplementary Figs. 11–13, 15 and 16 are available online[29,51,56,57,76,99]. Archive sediment core material is deposited at the Geological Survey of Denmark and Greenland, Aarhus University, and Stockholm University and available upon reasonable request by contacting Sofia Ribeiro (sri@geus.dk).

## Code availability

A detailed description of the processing of raw sequencing files, metadata files, taxonomic count data for DNA detections (shotgun sequencing and hybridization capture) and foraminifera assemblages used in this study are available at https://github.com/slennart/HHA-sedaDNA[128].

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

## Acknowledgements

We thank the captains, crew and scientific parties of NorthGreen 2017 Expedition onboard R/V *Dana*, ICAROS 2021 Expedition onboard HDMS *Lauge Koch*, and Ryder 2019 Expedition onboard R/V *Oden* (special thanks to Martin Jakobsson). Further, Christine Rømer is acknowledged for contributing to the micropalaeontological analysis. We also thank Mikkel Winther Pedersen and Benjamin Vernot for helpful discussions regarding the data analysis and the laboratory leaders and technicians of the Globe Institute, University of Copenhagen. This study was supported by the Villum Fonden Young Investigator Program (YIP +) grant no. 37352, Independent Research Fund Denmark (DFF) grant no. 9064-00025B, and the EliteForsk Prize to EDL; DFF grant no. 9064-00039B to SR; DFF grant no. 0135-00165B to MSS. The study also received funding from the Danish Center for Marine Research (DCH) and the European Union's Horizon 2020 research and innovation program under grant agreement No. 846142 (POLARC) to RJ, and the Horizon Europe program under grant agreement No. 101136480 (SEA-Quester) to MSS.

## Author contributions

L.S.: conceptualization, formal analysis, visualization, writing (original draft); S.R.: conceptualization, funding acquisition, resources, supervision, visualization, writing (review & editing); R.J.: investigation, formal analyzes, writing (review and editing); A.B.K.: investigation, formal analysis, writing (review & editing); K.N.: investigation, formal analysis, writing (review & editing); M.O.'R.: investigation, formal analysis, resources, writing (review & editing); C.P.: investigation, resources, writing (review & editing); F.S.: investigation, formal analysis, writing (review & editing); M.S.S.: funding acquisition, resources, writing (review & editing); H.H.Z.: investigation, methodology, writing (review & editing); E.D.L.: conceptualization, funding acquisition, resources, supervision, visualization, writing (review & editing).

## Competing interests

The authors declare no competing interests.
