## [Transparent Peer Review file · Nature Communications]

Holocene shifts in marine mammal distributions around Northern Greenland revealed by sedimentary ancient DNA

Corresponding Author: Mr Lennart Schreiber

Version 0:

Reviewer comments:

Reviewer #1

(Remarks to the Author)

The paper is about the distribution of marine mammals in northern Greenland during the Holocene. To this end, sediment cores from four different locations were examined and the occurrence of various species was detected using aDNA analysis. In addition, further environmental information from proxy investigations of the same sediments was used to link the distribution of the animals with the prevailing environmental conditions. The result significantly expands our previous knowledge of the animals' distribution, as the fossil record from coastal sites is patchy. I consider this work to be highly relevant and innovative because it combines biological techniques and geological archives and research questions.

For this review the focus is on the chronologies of the sediment records.

The age-depth models of the four cores are based on radiocarbon ages. Two of the models were taken from existing publications, one was created by correlation with a dated core, and the other was generated using newly obtained 14C ages.

The age models differ in their data basis (number of age constraints) and the cores differ in their temporal resolution. At this point, I highlight the core from Hall Basin, whose age model seems to be the weakest. Therefore, it is important to provide a comprehensive description of the approach and to give a detailed discussion on the limitations of the age-depth model of the core. This is not properly elaborated in the manuscript at the moment and should be added/modified. Although the correlation to a dated neighboring core is plausible, there is no dating for the base of the core, whose age is only estimated. This should be explained in more detail at some point. Furthermore, the last c. 7000 years seem to be very condensed in the record, or it could also be that the core is incomplete. The difficulty in estimating the age of this core (and of sediment cores from polar regions in general) cannot be avoided, but it must be explained more clearly and taken into account when interpreting the aDNA and other proxy data in terms of "age". I think, this is widely taken into account in the discussion, but it is not worked out clearly enough in the results/methods. More detailed comments and specific suggestions are given in the commented version of the main text and the supplement that I provide with this review.

(Remarks on code availability)

Reviewer #2

(Remarks to the Author)

This paper uses shotgun sequencing and hybridization capture to identify marine mammals from 4 cores around the northern shelf edges of the Greenland ice-sheet and land-mass. The results do indeed provide both a remarkable marine palaeo-record and also provide novel insights into the Lateglacial-Holocene dynamics of the Arctic and north Atlantic ocean systems around Greenland. This research area is critically important for studying future changes in these currents and their effects on marine resources in the face of climate change. This paper is also important in two other respects; firstly, it is one of very few and first long-term sedaDNA studies of marine systems and secondly it integrates marine mammals with other proxies, in this case forams. The methodology is sound and my only significant criticisms revolve around some more transparency of the data sources and some improvement in clarity of discussion. Major points for the authors consideration:

1. The damage patterns from the shotgun sequencing - are both quite low for their age and also show considerable scatter with a weak relationship to depth/age.. a bit more mention or discussion of this would be an improvement --and since these are means - on how many sequences are they based and could this be added to the diagram (there are some different ways of showing this data).
2. The dating is critical and not too bad for most of the cores but there are problems e.g. Hall Basin has a top problem - and it effects the 'timing' of the detections - what happens to the narrative if the upper 4 samples are all clustered between c. 7 ka and 4 ka as might be the case if this is an eroding site?
3. Shotgun.. was it not possible to take the eukaryote families such as Phocidae using standard reference libraries? if not why not? Any fragment length data?
4. This leads to the question of the sedimentological environment of each cores (not just depth and location) - Lincoln Sea core seems off the mouth of a fjord system. If not space in this paper can the reader be directed to papers on these cores if they exist (which is not clear).
5. It is as clear from the above what has been done on these cores before and what for this study?
6. Figs - whilst OK - it would be vary much better to try and combine the hybridization capture DNA records records together so the reader can see how the changes in the text effect the whole system?
Since the word count is over 2000 I think you can have more than 4 figures - I think the paper warrents at least 6 figures..
7. The discussion pages 9-17 seems a bit rambling and would be really helped also by a figure as suggested above.
8. The discussion on detection and preservation is a bit weak - what mineralogy of the sediments and did this vary - it is not clear from the data and this refers back to the sedimentology and sediment inputs.
9. Lastly, a new proxy is mentioned - IP25 of algal productivity - and although a reference given this needs more explanation here.

The manuscript is clean and well written.

(Remarks on code availability)

Reviewer #3

(Remarks to the Author)

This work presents an overview of ~12 thousand years of marine mammal habitation in northern Greenland. It notes the appearance of some taxa in the sedaDNA record much earlier than the palaeontological record. The authors also try to draw inferences on the abundance (or lack thereof) of certain taxa across this time period. The results are interesting, and I have no large problems with the methodology (at least as it pertains to the ancient DNA – I cannot speak to the core age modeling), and again serve to highlight the usefulness of sedaDNA as a complement to traditional palaeontological analyses.

I think the authors do a good job with the interpretation of their results when it comes to the identification of which taxa are present, and this is where I think the paper is strongest. However, I feel the argument suffers a bit when attempting to draw conclusions based on the absence of species, particularly when the presence of those taxa in identified bins is also quite low. This is most problematic in the discussion section titled "Episodic marine mammal detections in north-east Greenland over the past 10 ka", for which I've detailed my concerns below.

I'm not entirely sure I'm seeing the link the authors want me to see between harp seal presence/absence and primary productivity. While I again don't question the detection of taxa from this core, the authors also mention that the detection is quite poor (<10 reads). However, to elucidate a pattern of appearance/disappearance requires that periods of presence and absence are strongly separated, and when the presences are so weak it leads doubt to whether something is truly there and not being detected or actually absent. I suspect part of this is due to the extraction kits the authors chose to use – the PowerSoil kits have poor DNA recovery for ancient remains, although they generally manage to capture most of the most abundant taxa; see for example the comparison in Murchie et al. (2020; <https://doi.org/10.1017/qua.2020.59>). I would recommend swapping to a more sensitive extraction methodology for future work.

I'm also not entirely sure I'm correlating the shifts within the graphs in Fig. 4A and B with the harp seal abundance. Part of this may be my own unfamiliarity with benchmarks for these metrics (i.e. are there certain thresholds below which harp seals are never found?), but I'm not sure I see meaningfully different levels when harp seals are present or not. For example, the harp seal signature ends at ~ 7 ka BP right when 4B appears to show a local maxima, and levels similar to when harp seals are present continue for ~1.5 ky after they disappear. I think this section needs some caveats but would also benefit with better explanations that guide the reader through the correlations, especially as this journal has a broad scope.

Additional points:

- Line 131: I had a hard time linking samples/cores back to the Data Tables. For example, I'm wondering why data table 2

only has 66 libraries instead of the 109 mentioned here.

- Line 155 and Fig. S1: I don't quite understand what exactly is being shown in Fig. S1 and/or what the authors are trying to convey displaying the data this way. The main text appears to indicate the deamination patterns are split by taxa, but it is not discernable which taxa is which (or why this might be relevant) from the figure. I would at least supplement this with one more supplementary figure showing the mapped data for their key taxa (lumping across time points if necessary) and running it through MapDamage 2.0 to produce full deamination plots. I think a sentence or two in the methods explaining how exactly the deamination of these final 3bp was calculated wouldn't be remiss.

- Line 188: It is unclear to me how these numbers relate to Data Table 1. For example, the first library eDNALib006_050 has more than the upper limit mentioned here at 8133 reads assigned across Prokaryota and Eukaryota. The average for the libraries is also higher.

- Line 189: Also, not sure as to the source of these numbers. The lowest amount assigned to prokaryotes appears to be 64% in eDNALib006_016. I suspect this may be a relic from an earlier analysis?

- Line 215: Why is this window so broad? The sample ages spanning for most whales don't appear to produce such a large window. And if it's to include the seals harp seals (listed in this sentence), they appear to be detected from samples ~1 ka BP.

- Line 231: There should be a figure callout here as this starts a new section.

- Line 235: I'm having difficulty linking the cores and samples mentioned here to ones in the Data Tables. It would be great if they could have the same name. I believe the two for this are the ones labelled Ryder19_12_GC and Ryder19_24_PC.

- Line 247: This should specifically reference Fig. 4E.

- Line 275: I'm not sure the evidence for this claim comes across super strong by this point in the paper. It would be better removing this or moving it to the conclusion.

- Line 327: Latitude limits are mentioned here and in other parts of the paper. If not too much work (depending on how the map was made) it might be worth adding latitude lines to the Greenland blow-up in Fig. 1.

- Line 413: This shouldn't be capitalized as I assume this is not referring to the time period known as the Late Pleistocene (the last interglacial to the end of the Pleistocene), but instead the end of the Pleistocene. Maybe rephrase to something like "Towards the end of the Pleistocene ..."

- Line 532: This sentence serves no purpose and should be excluded. Or it needs to be caveated with why northernmost is a meaningful/interesting metric in the same way as oldest might be.

- Line 681: This should specify that you are removing mapped sequences as the previous sentences are about your reference database construction.

- Line 766: Raw sequencing data should always be uploaded to NCBI to allow for reanalysis in the future as better bioinformatic tools and reference databases become available.

(Remarks on code availability)

Github page is nice and well organized. Good code commenting.

Version 1:

Reviewer comments:

Reviewer #1

(Remarks to the Author)

The paper is about the temporal and spatial distribution of marine mammals in Greenland. The use of ancient DNA analyses on sediment cores from different regions has provided new insights into the timing of the first appearance of specific species and on the prevailing environmental conditions that may explain the pattern of species occurrence throughout the Holocene.

The chronology of the investigated sediment cores is fundamental for the interpretation of the temporal changes and correlation between regions/records. The authors used the widely accepted method of ^{14}C dating for this purpose. In the original manuscript version, I had noted that in a few places a more detailed description of the procedures of reservoir corrections, age-depth modelling and correlations was necessary to make the quality of the age information more assessable to the reader. The authors responded to this and fully improved the points mentioned.

Reviewer #2

(Remarks to the Author)

Reviewer #3

(Remarks to the Author)

My revisions have been addressed adequately. I have no further revisions.

REVIEWER COMMENTS

Reviewer #1 (Remarks to the Author):

The paper is about the distribution of marine mammals in northern Greenland during the Holocene. To this end, sediment cores from four different locations were examined and the occurrence of various species was detected using aDNA analysis. In addition, further environmental information from proxy investigations of the same sediments was used to link the distribution of the animals with the prevailing environmental conditions. The result significantly expands our previous knowledge of the animals' distribution, as the fossil record from coastal sites is patchy. I consider this work to be highly relevant and innovative because it combines biological techniques and geological archives and research questions.

For this review the focus is on the chronologies of the sediment records.

The age-depth models of the four cores are based on radiocarbon ages. Two of the models were taken from existing publications, one was created by correlation with a dated core, and the other was generated using newly obtained ^{14}C ages. The age models differ in their data basis (number of age constraints) and the cores differ in their temporal resolution.

At this point, I highlight the core from Hall Basin, whose age model seems to be the weakest. Therefore, it is important to provide a comprehensive description of the approach and to give a detailed discussion on the limitations of the age-depth model of the core. This is not properly elaborated in the manuscript at the moment and should be added/modified. Although the correlation to a dated neighboring core is plausible, there is no dating for the base of the core, whose age is only estimated. This should be explained in more detail at some point. Furthermore, the last c. 7000 years seem to be very condensed in the record, or it could also be that the core is incomplete. The difficulty in estimating the age of this core (and of sediment cores from polar regions in general) cannot be avoided, but it must be explained more clearly and taken into account when interpreting the aDNA and other proxy data in terms of "age". I think this is widely taken into account in the discussion, but it is not worked out clearly enough in the results/methods. More detailed comments and specific suggestions are given in the commented version of the main text and the supplement that I provide with this review.

Response: We now provide a much more comprehensive description of the age model for the Hall Basin core and how it was derived (L601-621). This includes how the age for the transition into the basal diamict was derived – which is consistent with previous work on this site (Jakobsson et al., 2018, Jennings et al., 2022). An important fact that we now explicitly highlight in the manuscript is that the drop in sedimentation rates for samples younger than 8-7 ka is not something unique to Ryder19-24PC. In fact this same pattern is seen across the Nares Strait and Hall Basin, as illustrated by Jennings et al., 2022, and north of the Robeson Channel in the Lincoln Sea (Detlef et al., 2023). There are no indications of hiatus in the core, and the changes in sedimentation rate are linked to changes in ice shelf sedimentation processes. With regards to our earliest DNA detections it is important to highlight that all marine mammal detections were found in the sediment layers above the deepest radiocarbon age, so our ecological inferences are not affected by the estimated age for the top of the diamict.

Specific comments:

1. x-Axes labelling: Please use Age (cal ka BP). Ka = kilo year
Response: We have adjusted the x-axes labelling for all figures generated for this study to 'Age (cal ka BP)' as suggested.
2. Species names in panel b in italics; Note formatting in panel b: 13 superscript
Response: We have changed the font to italics for latin species names and adjusted the spelling of $\delta^{13}\text{C}$ as suggested.
3. What is the age range you found? Give reference to the ^{14}C data table and mention results in text.
Response: We have added a reference to the ^{14}C data table (Table S2) and mention the calibrated radiocarbon dates in the text (L167-169).
4. What did you use as an age constraint?
Response: We have added information on the upper age constraint (year = 2021 CE), which relates the surface of the sediment core to the year of retrieval; the core was collected in 2021 (L171).
5. Add to which age this depth corresponds
Response: We have added the calibrated age of the depth where the sedimentology changes: 213 cm depth corresponds to 12.3 ± 1.4 cal ka BP (L179).
6. Add reference to Figure S4 and Table S2 and S1 (presumably)
Response: Done.
7. Give core ID in table caption
Response: Done.
8. Please check the axis labelling: probably "cal yr BP" and not "cal ka BP"?
Response: The x-axis title did not match the x-axis labels in the previous version of Fig. S10. In the new version of the figure, the x-axis title and x-axis labels match.
9. Please report original ^{14}C age (as given in the publication by Bennike et al 1989) and calibrate the age. Calibration should be performed with marine20 and with and appropriate DR.
Response: We have created a new table in the supplement (Table S7), where we present the original fossil ^{14}C ages, reservoir offsets, and newly calibrated ages (using Marine20).
10. This is a description of the method and not of the results. It is important to know how many age tiepoints were used, what ages they have and what the age model looks like. For example, the mid and late Holocene is not well resolved in this core either. The last dating point is 6900 cal yr BP. Was it also ensured for this core that the core top is the modern sediment surface and therefore a low sedimentation rate can be assumed, or is it possible that parts of the core are missing? Please add these aspects to the description. It is important for the discussion to point out the weaknesses of the age assignment in this record.
Response: As further detailed above, we have revised the age model for the Hall Basin core and provided a more comprehensive description in the Methods section (L601-621). We have also revised the respective description in the Results section, including the same details and structure as for the Melville Bay core (L187-205). Reduced sedimentation rates younger than 8-7 ka are a robust feature in all records from the Nares Strait region. And thus, we do not think that this site suffers from local erosion. We agree that this adds some uncertainty

to the exact timing of events in the Mid-to-Late Holocene for samples with estimated ages younger than the oldest radiocarbon date (6.9 ± 0.1 cal ka BP). Therefore, our discussion of marine mammal detections for both Hall Basin 24PC and Lincoln Sea 12-GC focuses mainly on the earliest detection of marine mammals in the region (L439-484).

11. Why did you use marine13 and not marine20 (as for the core from Melville Bay)?

Response: We have changed this now and re-calibrated all ages using Marine20. Originally, we adopted the reported dates from Jennings et al., 2022 – and simply wanted to be consistent. After a required adjustment to the local reservoir correction (subtracting 150 years to translate dR13 to dR20) we generated a new age model using *OxCal*. There is very little difference between these age models, except for a 400-year difference at the base of the sequence, where the calibrated age for the top of the diamict is now younger (see Figure 1 below). The reason for this difference is that in the original submission we assumed a basal age of 9600 cal yrs BP, which was based on the calibrated age (using Marine 13) for the ice margin retreat provided by Jakobsson et al. 2018. However, this was the oldest estimate they reported and was derived by assuming no reservoir correction. In our revised Hall Basin 24PC chronology, the uncalibrated age for the ice margin retreat is basically the same as for the deepest benthic radiocarbon date in the record. It is apparent that the laminated sediments were very rapidly deposited. Without direct age constraints, the exact timing for the deposition of the laminated sequence remains uncertain. However, this part of the core did not contain any positive reads for marine mammal DNA and thus it doesn't impact the environmental interpretations of this manuscript.

Figure 1. Comparison of the age model used in our first submission and the revision, after we have addressed the reviewer's request to update the calibration (and hence the dR) using Marine20, and adjusting the age for the top of the diamicton.

12. Why do you use a value of 510 years for DR? It is much higher than values reported for northern Greenland (DOI:10.5194/gchron-5-451-2023) Please add some discussion on this.
Response: A dR of 510 years for benthics (using Marine 13) is consistent with recent literature for other marine sediment cores from Hall Basin (doi.org/10.1016/j.quascirev.2022.107460). We have adjusted the dR for calibration with Marine20 (L607-610).
13. Explain briefly why you set the onset of marine deposition at this location to 9600 years BP
Response: We have added this explanation to the text (L610-616).
14. Why do you use clam for this core and BACON for the other?
Response: There was no particular reason. We have now used *OxCal* to model the ages, so all reported models are based on Bayesian statistics. .
15. Please structure the section in the same way as for the Melville Bay core: number of 14C ages, number and definition of age constraints other than 14C ages, which calibration did you use and which DR.
Response: Thank you, we have re-structured the respective section according to this suggestion (L187-205).
16. The age estimate of the base of the record is not derived from 14C age (as indicated in data table 4) and should differ graphically from the calibrated 14C ages.
Response: We have updated the chronology of Hall Basin 24PC (Fig. S10) and the age estimate of the base of the record now differs graphically from the ¹⁴C ages.
17. I think the presentation of the age ranges for each sample is a good idea. However, the way it is presented is unsuitable for the upper part of the Hall Basin core. The bars suggest that the age assessment for the sample points is very precise, but in fact it is purely speculative.
Response: We have argued that the low sedimentation rates between the uppermost radiocarbon date at Hall Basin 24PC (circa 6900 14C years) and the surface of the core (about 4.3 cm/ka) is very much consistent with other records from the Hall Basin and Lincoln Sea (i.e the region around the Nares Strait). We do not think there is evidence for local on-deposition here. We accept that there is uncertainty beyond the reported calibration and modelling uncertainties, however this is true for all age models. We can arbitrarily increase the uncertainty by changing the potential error of the reservoir correction - but our justification for not doing this is to remain consistent with previous work in the area (e.g. Jakobsson et al. 2018; Jennings et al. 2022; Detlef et al. 2023) .
18. Reporting of 14C ages: Give F14C values and apply rounding convention of reported 14C ages. What is meant by “modeled ages”- if you refer to calibration, please use the terms 14C ages and calibrated ages instead.
Response: We have updated Table S2 by adding F¹⁴C values in a new column, rounded ages to the closest decade, and adjusted the table legend as suggested.
19. This interpretation should be made with caution: the oldest foraminifer age obtained from the core is 9.3-10.9 ka, so it can be stated that the first occurrence of marine mammal DNA pre-dates this. Is it plausible to expect the presence of marine mammals at this site prior to ice sheet retreat?
Response: We have adjusted the reporting of the estimated age of this particular sample to be ≥11.6 - 11.0 cal ka BP (L231). This is more consistent with our interpretation that samples of this core with depths ≥213 cm were likely deposited at or just prior to 11.6 ka BP. While there is no further age constraint for these samples, we cannot exclude the possibility of

marine mammal DNA deposition in the sediment under a marine-based ice shelf, prior to the ice-sheet retreat recorded on land.

Reviewer #2 (Remarks to the Author):

This paper uses shotgun sequencing and hybridization capture to identify marine mammals from 4 cores around the northern shelf edges of the Greenland ice-sheet and land-mass. The results do indeed provide both a remarkable marine palaeo-record and also provide novel insights into the Lateglacial-Holocene dynamics of the Arctic and north Atlantic ocean systems around Greenland. This research area is critically important for studying future changes in these currents and their effects on marine resources in the face of climate change. This paper is also important in two other respects; firstly, it is one of very few and first long-term sedaDNA studies of marine systems and secondly it integrates marine mammals with other proxies, in this case forams. The methodology is sound and my only significant criticisms revolve around some more transparency of the data sources and some improvement in clarity of discussion. Major points for the authors consideration:

1. The damage patterns from the shotgun sequencing - are both quite low for their age and also show considerable scatter with a weak relationship to depth/age.. a bit more mention or discussion of this would be an improvement --and since these are means - on how many sequences are they based and could this be added to the diagram (there are some different ways of showing this data).

Response: We have added four new figures to the supplement (Fig. S2 - S5) showing damage patterns for DNA sequences assigned to Phocidae (seals) across the four marine sediment cores, including fragment length distributions and coverage estimates (L778).

2. The dating is critical and not too bad for most of the cores but there are problems e.g. Hall Basin has a top problem - and it effects the 'timing' of the detections - what happens to the narrative if the upper 4 samples are all clustered between c. 7 ka and 4 ka as might be the case if this is an eroding site?

Response: This is a fair observation that we have addressed in the revised manuscript (L601-621). We have better described how the age model for this core was constructed and added a facies interpretation to go along with the correlation of physical and chemical proxies (MSCL and XRF). An important fact that we now highlight in the manuscript is that the reduction in sedimentation rates younger than 8-7 ka is not something unique to Ryder19-24PC. In fact this same pattern is seen across the Nares Strait and Hall Basin, as illustrated by Jennings et al., 2022, and north of the Robeson Channel in the Lincoln Sea (Detlef et al., 2023). There is no reason to assume that this core suffers from erosion of near-surface sediments.

3. Shotgun.. was it not possible to take the eukaryote families such as Phocidae using standard reference libraries? if not why not? Any fragment length data?

Response: Using shallow shotgun sequencing, we anticipated not recovering the complete eukaryote community (L212-216). We refrained from deeper shotgun sequencing due to the high costs of generating enough data for recovering rare DNA sequences (as expected for marine

mammal DNA). We have added fragment length data (Fig. S2 - S5) for the hybridization capture data assigned to Phocidae.

4. This leads to the question of the sedimentological environment of each cores (not just depth and location) - Lincoln Sea core seems off the mouth of a fjord system. If not space in this paper can the reader be directed to papers on these cores if they exist (which is not clear).

Response: We have now carefully checked, and all relevant papers are referenced for the cores (L 586-630; L785-796). For Melville Bay 26G, a detailed description of the sedimentological environment is currently being prepared for submission. For Lincoln Sea 12GC, we added a previously missing reference (Cronin et al. 2022) describing the oceanographic conditions and glacial history of Southern Lincoln Sea (L452).

5. It is as clear from the above what has been done on these cores before and what for this study?

Response: In the methods, we include a section titled "Paleoenvironmental and paleoclimatic data for comparison". This section provides details on previously published paleoenvironmental data from the same or nearby sediment cores and newly generated data as part of this study (L785-796).

6. Figs - whilst OK - it would be vary much better to try and combine the hybridization capture DNA records records together so the reader can see how the changes in the text effect the whole system? Since the word count is over 2000 I think you can have more than 4 figures - I think the paper warrants at least 6 figures..

Response: We now include a figure where the hybridization capture data are presented for each marine mammal species, across cores (Fig S14). It is in the SI, to avoid presenting the same findings twice in the main text.

7. The discussion pages 9-17 seems a bit rambling and would be really helped also by a figure as suggested above.

Response: We have added the requested figure to the SI, which we hope will aid the reader. We have revised sections of the discussion to make it more succinct (L508-525). However, there are so many exciting findings and inferences to highlight, that the discussion still has some length.

8. The discussion on detection and preservation is a bit weak - what mineralogy of the sediments and did this vary - it is not clear from the data and this refers back to the sedimentology and sediment inputs.

Response: We acknowledge the reviewer's point. This is an emerging field within sedaDNA studies and we have initiated XRD (X-ray diffraction) analyses of marine sediments from the region for another ongoing study where we aim to reconstruct the entire ecosystem going further back in time. However, a detailed mineralogical analysis of these Holocene records is beyond the scope of this study, where we followed a targeted approach (hybridization capture of marine mammals).

9. Lastly, a new proxy is mentioned - IP25 of algal productivity - and although a reference given this needs more explanation here.

Response: We have included an explanation on the use of IP25 in the context of paleoceanographic reconstructions (L366-368).

The manuscript is clean and well written.

Reviewer #3 (Remarks to the Author):

This work presents an overview of ~12 thousand years of marine mammal habitation in northern Greenland. It notes the appearance of some taxa in the sedaDNA record much earlier than the palaeontological record. The authors also try to draw inferences on the abundance (or lack thereof) of certain taxa across this time period. The results are interesting, and I have no large problems with the methodology (at least as it pertains to the ancient DNA – I cannot speak to the core age modeling), and again serve to highlight the usefulness of sedaDNA as a complement to traditional palaeontological analyses.

I think the authors do a good job with the interpretation of their results when it comes to the identification of which taxa are present, and this is where I think the paper is strongest. However, I feel the argument suffers a bit when attempting to draw conclusions based on the absence of species, particularly when the presence of those taxa in identified bins is also quite low. This is most problematic in the discussion section titled “Episodic marine mammal detections in north-east Greenland over the past 10 ka”, for which I’ve detailed my concerns below.

I’m not entirely sure I’m seeing the link the authors want me to see between harp seal presence/absence and primary productivity. While I again don’t question the detection of taxa from this core, the authors also mention that the detection is quite poor (<10 reads). However, to elucidate a pattern of appearance/disappearance requires that periods of presence and absence are strongly separated, and when the presences are so weak it leads doubt to whether something is truly there and not being detected or actually absent. I suspect part of this is due to the extraction kits the authors chose to use – the PowerSoil kits have poor DNA recovery for ancient remains, although they generally manage to capture most of the most abundant taxa; see for example the comparison in Murchie et al. (2020; <https://doi.org/10.1017/qua.2020.59>). I would recommend swapping to a more sensitive extraction methodology for future work.

Response: We renamed this section of the discussion (L502) and toned down the interpretations, especially the correlation of sparse detections with paleoceanographic reconstructions (L508-525). The discussion now focuses more on broad detection patterns and acknowledges the overall sparsity of marine mammal DNA in this record. We will consider using an alternative, more efficient extraction protocol for future laboratory work on marine sediment cores.

I’m also not entirely sure I’m correlating the shifts within the graphs in Fig. 4A and B with the harp seal abundance. Part of this may be my own unfamiliarity with benchmarks for these metrics (i.e. are there certain thresholds below which harp seals are never found?), but I’m not sure I see meaningfully different levels when harp seals are present or not. For example, the harp seal signature ends at ~ 7 ka BP right when 4B appears to show a local maxima, and levels similar to when harp seals are present continue for ~1.5 ky after they disappear. I think this section needs some caveats but would also benefit with better explanations that guide the reader through the correlations, especially as this journal has a broad scope.

Response: We revised this section of the discussion (L508-525) as described for the previous comment.

Additional points:

- Line 131: I had a hard time linking samples/cores back to the Data Tables. For example, I'm wondering why data table 2 only has 66 libraries instead of the 109 mentioned here.

Response: The data tables in our original submission only contained samples and taxonomic assignments containing at least 3 unique sequences (hybridization capture) or 10 unique sequences (shotgun sequencing). We have revised the tables containing the taxonomic assignments of shotgun sequencing and hybridization capture. Table S1 (shotgun sequencing) contains all 55 samples & ten blanks; Table S9 (hybridization capture) now contains all 116 samples, and 25 blanks.

- Line 155 and Fig. S1: I don't quite understand what exactly is being shown in Fig. S1 and/or what the authors are trying to convey displaying the data this way. The main text appears to indicate the deamination patterns are split by taxa, but it is not discernable which taxa is which (or why this might be relevant) from the figure. I would at least supplement this with one more supplementary figure showing the mapped data for their key taxa (lumping across time points if necessary) and running it through MapDamage 2.0 to produce full deamination plots. I think a sentence or two in the methods explaining how exactly the deamination of these final 3bp was calculated wouldn't be remiss.

Response: We have added four new figures to the supplement (Fig. S2 - S5) showing damage patterns for DNA sequences assigned to Phocidae across the four marine sediment cores, including fragment length distributions and coverage estimates (L778). We have also added a more detailed description about the data underlying Fig. S1 in the methods section (L769-778).

- Line 188: It is unclear to me how these numbers relate to Data Table 1. For example, the first library eDNALib006_050 has more than the upper limit mentioned here at 8133 reads assigned across Prokaryota and Eukaryota. The average for the libraries is also higher.

Response: The reason for the mismatch between the numbers in the main text and the numbers in the table stems from the table only listing taxonomic assignments above the threshold of 10 reads, while the numbers in the text were based on all taxonomic assignments, including those with less than 10 reads. We have revised the numbers in the main text to match the numbers in the table (L208-210).

- Line 189: Also, not sure as to the source of these numbers. The lowest amount assigned to prokaryotes appears to be 64% in eDNALib006_016. I suspect this may be a relic from an earlier analysis?

Response: We have revised the numbers, as stated in the previous response.

- Line 215: Why is this window so broad? The sample ages spanning for most whales don't appear to produce such a large window. And if it's to include the seals harp seals (listed in this sentence), they appear to be detected from samples ~1 ka BP.

Response: We adjusted the window to 8 - 7 cal ka BP to account for the fact that most of the listed species were only detected in that period (L237).

- Line 231: There should be a figure callout here as this starts a new section.

Response: We added a figure reference (Fig. 3e) at the end of the sentence (L474).

- Line 235: I'm having difficulty linking the cores and samples mentioned here to ones in the Data Tables. It would be great if they could have the same name. I believe the two for this are the ones labelled Ryder19_12_GC and Ryder19_24_PC.

Response: We have adjusted the core names in the Data Tables (now called Table S1 & S9) so they match the main text.

- Line 247: This should specifically reference Fig. 4E.

Response: We added the figure reference at the end of the sentence (L523).

- Line 275: I'm not sure the evidence for this claim comes across super strong by this point in the paper. It would be better removing this or moving it to the conclusion.

Response: We have removed the sentence.

- Line 327: Latitude limits are mentioned here and in other parts of the paper. If not too much work (depending on how the map was made) it might be worth adding latitude lines to the Greenland blow-up in Fig. 1.

Response: Thank you, we have added latitude lines to Fig. 1 to make cross-referencing of the latitudes mentioned in the main text easier.

- Line 413: This shouldn't be capitalized as I assume this is not referring to the time period known as the Late Pleistocene (the last interglacial to the end of the Pleistocene), but instead the end of the Pleistocene. Maybe rephrase to something like "Towards the end of the Pleistocene ..."

Response: Thank you, we have adopted the proposed phrasing (L443).

- Line 532: This sentence serves no purpose and should be excluded. Or it needs to be caveated with why northernmost is a meaningful/interesting metric in the same way as oldest might be.

Response: We have removed the sentence.

- Line 681: This should specify that you are removing mapped sequences as the previous sentences are about your reference database construction.

Response: We have rephrased the sentence for clarification (L739-740).

- Line 766: Raw sequencing data should always be uploaded to NCBI to allow for reanalysis in the future as better bioinformatic tools and reference databases become available.

Response: We have created a SRA BioProject (PRJNA1211513) with the raw sequencing files for the metagenomic shotgun sequencing data and hybridization capture sequencing data. The data availability was updated with the SRA BioProject reference (L830-831). The project will be made publicly available upon publication.

Reviewer #3 (Remarks on code availability):

Github page is nice and well organized. Good code commenting.